# A Scalable Inter-edge Correlation Modeling in CopulaGNN for Link Sign Prediction

**Jinkyu Sung, Myunggeum Jee, Joonseok Lee**[*]
Seoul National University
{jinkyusung, goldmango, joonseok}@snu.ac.kr

## Abstract

Link sign prediction on a signed graph is a task to determine whether the relationship represented by an edge is positive or negative. Since the presence of negative edges violates the graph homophily assumption that adjacent nodes are similar, regular graph methods have not been applicable without auxiliary structures to handle them. We aim to directly model the latent statistical dependency among edges with the Gaussian copula and its corresponding correlation matrix, extending CopulaGNN (Ma et al., 2021). However, a naive modeling of edge-edge relations is computationally intractable even for a graph with moderate scale. To address this, we propose to 1) represent the correlation matrix as a Gramian of edge embeddings, significantly reducing the number of parameters, and 2) reformulate the conditional probability distribution to dramatically reduce the inference cost. We theoretically verify scalability of our method by proving its linear convergence. Also, our extensive experiments demonstrate that it achieves significantly faster convergence than baselines, maintaining competitive prediction performance to the state-of-the-art models.

## 1 Introduction

A signed graph refers to a graph where edges can be either positive ($+$) or negative ($-$). In the real world, a variety of systems containing positive and negative relationship between entities can be modeled as a signed graph; *e.g.*, social networks (friendly/hostile) or content recommendation (like/dislike) (Liao et al., 2018; Kim et al., 2024; Chen et al., 2024; Kim et al., 2025b). On a signed graph, we are interested in the *link sign prediction* task, aiming at predicting the sign of unobserved edges, given the partially observed positive and negative edges in a graph (Agrawal et al., 2013).

In spite of recent advances in general graph neural networks (GNNs) (Kipf & Welling, 2017; Gilmer et al., 2017), they are not directly applicable to the link sign prediction problem, because the presence of negative edges violates the traditional homophily assumption, implying that adjacent nodes are similar (Zhu et al., 2020). As the regular GNN methods cannot be simply adapted to this problem, researchers have proposed signed graph neural networks (SGNNs) to specially handle negative edges, *e.g.*, graph preprocessing inspired by sociological principles like structural balance and status theory (Derr et al., 2018; Huang et al., 2019; 2021) or separate treatment of negative edges (Li et al., 2020; 2023).

However, these additional auxiliary structures often lead to slow convergence, and in some cases, significantly inefficient use of the memory space. In contrast to adding auxiliary operations, we shift our attention to a more fundamental aspect: statistical *dependency among edges*. Instead of assuming that the adjacent nodes are similar, we assume that adjacent *edges* connected via a common node are *not informationally independent*; that is, they are either in accordance with or in the opposite to each other. This relaxation naturally extends the idea of GNNs from an unsigned graph to a signed one. Note that our assumption is edge-focused since we target the link sign prediction problem.

A similar perspective has been developed by CopulaGNN (Ma et al., 2021) for node regression tasks on unsigned graphs. By learning a Gaussian copula (Stute, 1986; Schweizer, 1991) and its corresponding correlation matrix, CopulaGNN aims at discovering the underlying dependency structure

---

[*]Corresponding author

among *nodes* in the graph. In this paper, we extend the node-centric CopulaGNN framework to the edge-centric task of link sign prediction. This extension, however, is not straightforward due to the following challenges. At training, unlike an unsigned graph where its inter-node correlations can be simply parameterized with a Graph Laplacian based on edge connectivity, a signed graph requires direct modeling of correlations between edges. However, since the correlation matrix quadratically scales with the number of edges, the memory consumption prohibitively grows up to $O(|\mathcal{V}|^4)$ where $\mathcal{V}$ is the set of nodes in the graph. Unless the target graph is sufficiently sparse, it is impractical to directly model this correlation matrix. At inference, CopulaGNN estimates and samples from the conditional probability distribution of the unobserved data given the observed one. This process includes inverting the huge correlation matrix, incurring prohibitively high computational cost.

To overcome these challenges, we propose **CopulaLSP** (**Copula**GNN for **L**ink **S**ign **P**rediction), a novel framework that enables efficient and scalable modeling of inter-edge correlations for link sign prediction. Our framework introduces two key components to achieve this scalability. At training, we construct the correlation matrix as a Gramian of edge embeddings. This novel strategy significantly reduces the number of learnable parameters, while still allowing the direct modeling of inter-edge dependencies. At inference, we reformulate the conditional probability distribution using the Woodbury matrix identity (Woodbury, 1950), which simplifies the inversion of a matrix with the Gramian structure, transforming the problem into the inversion of a considerably smaller matrix. Our proposed reformulation adapts the CopulaGNN to edge-centric tasks, resolving the prohibitive memory and computation costs that make the naive approach intractable.

Through extensive experiments, we demonstrate that our proposed method converges significantly faster than the baselines. We subsequently provide a theoretical foundation for this empirical observation, proving that our approach achieves linear convergence. This theoretical finding validates that the explicit and scalable modeling of inter-edge correlation is the key driver behind the dramatic acceleration in convergence speed.

## 2 PRELIMINARIES

**Problem Formulation.** Let $\mathcal{G} \coloneqq (\mathcal{V}, \mathcal{E})$ be a signed graph with a set of nodes $\mathcal{V}$ and edges $\mathcal{E}$. The objective of the semi-supervised link sign prediction task is to infer the missing signs for a set of unobserved edges. The set $\mathcal{E}$ of edges is partitioned into the observed $\mathcal{E}_{\mathrm{obs}}$ and the missing $\mathcal{E}_{\mathrm{miss}}$. Each edge $e_i \in \mathcal{E}_{\mathrm{obs}}$ has its known sign label $y_i \in \{-1, +1\}$, while it is unknown for the edges in $\mathcal{E}_{\mathrm{miss}}$. We denote the total number of edges by $n \coloneqq |\mathcal{E}|$ and the number of observed edges by $m \coloneqq |\mathcal{E}_{\mathrm{obs}}|$. For convenience, we assume all $n$ edges are ordered such that the observed ones $e_{1:m} \coloneqq (e_1, e_2, \ldots, e_m)$ come first, followed by the unobserved edges $e_{m+1:n} \coloneqq (e_{m+1}, e_{m+2}, \ldots, e_n)$.

**Sklar's Theorem.** According to *Sklar's theorem* (Sklar, 1959), any multivariate joint cumulative distribution function (CDF) $\mathcal{H}$ of $n$ univariate continuous random variables $X_{1:n}$ can be decomposed into its *marginal* CDFs $F_i(x) \coloneqq P(X_i \leq x)$ and *the copula function* $\mathcal{C} : [0,1]^n \to [0,1]$ by:

$$\mathcal{H}(x_{1:n}) = \mathcal{C}(F_1(x_1), F_2(x_2), \cdots, F_n(x_n)). \tag{1}$$

The copula function isolates and models the underlying dependence structure across the random variables, separating it from their individual marginal distributions.

Moreover, by *the probability integral transform* (David & Johnson, 1948), the transformed random variable $U_i \coloneqq F_i(X_i)$ follows the standard uniform distribution on [0, 1]. Then, the copula function $\mathcal{C}$ can be expressed in the following explicit form, not as a composite function of $F_i$:

$$\mathcal{C}(u_{1:n}) = \mathcal{H}(F_1^{-1}(u_1), F_2^{-1}(u_2), \cdots, F_n^{-1}(u_n)), \tag{2}$$

where $u_i \coloneqq F_i(x_i)$. If $\mathcal{H}$ is differentiable, its probability density function (PDF) $\mathcal{H}'$ is defined as:

$$\mathcal{H}'(x_{1:n}) \coloneqq \frac{\partial^n \mathcal{H}}{\partial x_1 \partial x_2 \cdots \partial x_n} = c(u_{1:n}) \prod_{i=1}^{n} f_i(x_i), \tag{3}$$

where $f_i$ is the PDF of $X_i$, and $c(u_{1:n}) \coloneqq \partial^n \mathcal{C} / \partial u_1 \partial u_2 \cdots \partial u_n$ is *the copula density*.

**Gaussian Copula.** We extend CopulaGNN (Ma et al., 2021) for the link sign prediction task, framed as binary classification of edge labels. While various copula functions can model the dependencies between variables, we specifically choose the *Gaussian copula* $\mathcal{C} : [0,1]^n \to [0,1]$ defined as:

$$\mathcal{C}(u_{1:n}; \mathbf{R}) \coloneqq \Phi_n(\Phi^{-1}(u_1), \Phi^{-1}(u_2), \cdots, \Phi^{-1}(u_n); \mathbf{0}, \mathbf{R}), \tag{4}$$

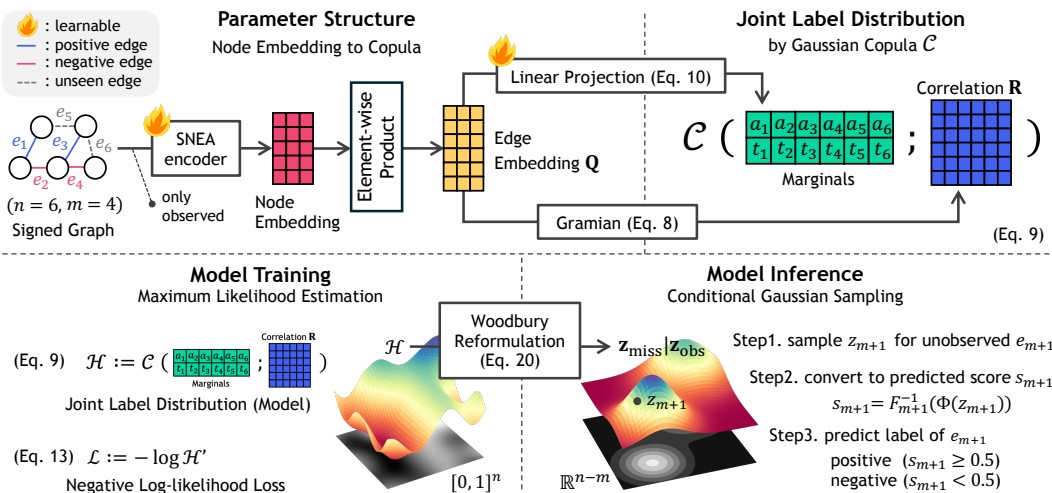

Figure 1: CopulaLSP (our proposed model) architecture and its training, inference process.

where $\Phi_n(\cdot; \mathbf{0}, \mathbf{R})$ is the multivariate Gaussian CDF with zero mean and its covariance is equivalent to the correlation matrix $\mathbf{R}$, and $\Phi^{-1}(\cdot)$ is the inverse CDF of the univariate standard Gaussian distribution. The *Gaussian copula density function* $c(\cdot; \mathbf{R})$ is naturally defined as:

$$c(u_{1:n}; \mathbf{R}) := \frac{\partial^n \mathcal{C}}{\partial u_1 \partial u_2 \cdots \partial u_n} = \frac{1}{\sqrt{\det \mathbf{R}}} \exp\left(-\frac{1}{2}\mathbf{z}^\top(\mathbf{R}^{-1} - \mathbf{I}_n)\mathbf{z}\right), \tag{5}$$

where $\mathbf{I}_n := \operatorname{diag}(1, \ldots, 1) \in \mathbb{R}^{n \times n}$, and $\mathbf{z} := (\Phi^{-1}(u_1), \Phi^{-1}(u_2), \cdots, \Phi^{-1}(u_n)) \in \mathbb{R}^n$.

This choice is motivated by its ability to naturally model the underlying dependency structure between edges through the correlation matrix, well-established by related studies (Jia & Benson, 2020; Ma et al., 2021).

## 3 METHODOLOGY

This section details our approach to effectively model the correlations between edges. We begin by defining the marginal probability distribution for each edge sign, followed by the introduction of our proposed correlation structure. As illustrated in Fig. 1, the marginal distributions and the correlation structure are then integrated using the Gaussian copula to form the joint probability distribution over all edge signs. Subsequently, we describe the training and inference procedures of our approach.

### 3.1 JOINT EDGE LABEL DISTRIBUTION AS LINK SIGN PREDICTION MODEL

**Edge Label Distribution.** To model inter-edge correlation with the Gaussian copula, we first define a marginal distribution $F$ for the sign of each edge. Recalling that we model the link sign prediction as a binary classification task, the edge signs can be intuitively modeled as a Bernoulli distribution. For the tractability of the joint PDF, we employ *the continuous relaxation* (Jang et al., 2017; Maddison et al., 2017) to convert the discrete Bernoulli distribution into a continuous one. The resulting PDF of *the relaxed Bernoulli distribution* is given by:

$$f(x; a, t) := \frac{atx^{-t-1}(1-x)^{-t-1}}{(ax^{-t} + (1-x)^{-t})^2} \quad \text{for } x \in (0, 1), \tag{6}$$

where $a$ and $t$ are the location and temperature parameters, respectively. In our model, $a \in (0, \infty)$ determines the sign of an edge, depending on whether it is greater or less than 1. The other parameter $t \in (0, 1)$ represents the confidence of the sign. Fig. 2 illustrates that when the confidence is high (top), the relaxed Bernoulli PDF shows higher density to the direction of the label (here, positive, to the right). On the other hand, when the confidence is low (bottom), the PDF exhibits non-negligible density not just for the label direction (negative, left), but for the opposite (positive, right).

Figure 2: Examples of relaxed Bernoulli marginals and a joint PDF using the Gaussian copula

In order to employ the Gaussian copula, we need the closed-form CDF $F(\cdot; a, t)$ and its inverse, which are derived as follows (see Appendix A):

$$F(x; a, t) = \frac{x^t}{a(1-x)^t + x^t}, \quad F^{-1}(x; a, t) = \frac{x^{1/t}}{a^{-1/t}(1-x)^{1/t} + x^{1/t}} \quad \text{for } x \in [0, 1]. \quad (7)$$

**Gramian of Edge Embedding as Edge Label Correlation.** To represent the dependency structure across the marginals, we require a correlation matrix $\mathbf{R} \in [-1, 1]^{n \times n}$. However, directly learning this $n \times n$ matrix $\mathbf{R}$ is memory-prohibitive. Moreover, $\mathbf{R}$ must be positive definite, a constraint required by the $\det \mathbf{R}$ term in the Gaussian copula density (Eq. (5)). To satisfy these two conditions, we construct the correlation matrix $\mathbf{R}$ from the Gramian of an edge embedding matrix $\mathbf{Q} \in \mathbb{R}^{n \times d}$:

$$\mathbf{R} := \nu(\mathbf{\Sigma}) = \mathbf{D}^{-1} \mathbf{\Sigma} \mathbf{D}^{-1}, \quad \text{such that} \quad \mathbf{\Sigma} := \mathbf{Q}\mathbf{Q}^\top + \epsilon \mathbf{I}_n, \quad (8)$$

where $d$ is the size of edge embeddings, $\epsilon$ is a hyperparameter in $(0, \infty)$, and $\nu(\cdot)$ normalizes a covariance matrix into a correlation matrix with a diagonal matrix $\mathbf{D}_{ii} := \sqrt{\mathbf{\Sigma}_{ii}}$. Recalling that a Gramian is only positive semi-definite, we ensure a covariance $\mathbf{\Sigma}$ becomes positive definite by adding a scaled identity term, $\epsilon \mathbf{I}_n$. Consequently, by *Sylvester's law of inertia* (Sylvester, 1852), $\mathbf{R}$ is also positive definite, with its minimum eigenvalue being positively lower-bounded.

We construct the edge embedding between two nodes using the element-wise product of the two node embeddings. The node embeddings are obtained from the signed graph using a signed graph encoder, *e.g.*, Derr et al. (2018); Li et al. (2020); Huang et al. (2021); Li et al. (2023).

Our approach is highly memory-efficient for two reasons. First, each edge is represented by a vector of a small size (*i.e.*, $d \ll n$). Second, the model learns only the parameters of the signed graph encoder, rather than the edge embeddings themselves. Additionally, we demonstrate that our approach has sufficient representational power to capture inter-edge correlation (see Appendix B).

**Overall Architecture.** We now have the Gaussian copula with relaxed Bernoulli marginals for edges and the learnable correlation matrix to model dependencies between them. Putting them together, our model is given by the following joint distribution:

$$\mathcal{H}(x_{1:n}; a_{1:n}, t_{1:n}, \mathbf{R}) = \mathcal{C}(F_1(x_1; a_1, t_1), F_2(x_2; a_2, t_2), \cdots, F_n(x_n; a_n, t_n); \mathbf{R}), \quad (9)$$

where $F_i(\cdot; a_i, t_i)$ is the relaxed Bernoulli CDF of edge $e_i \in \mathcal{E}$, and $\mathcal{C}(\cdot; \mathbf{R})$ is the Gaussian copula.

It is noteworthy that while the correlation matrix $\mathbf{R} \in [-1, 1]^{n \times n}$ is usually huge, its structure can be represented as the Gramian form thanks to the properties of the normalization function $\nu(\cdot)$. For this reason, the Gaussian copula $\mathcal{C}$ can be implemented with a low-rank multivariate Gaussian distribution without imposing a memory bottleneck (see Appendix C).

The parameters of the marginal distributions, $a_i$ and $t_i$, are obtained from the edge embeddings $\mathbf{Q}$ through two different learnable linear projections $\mathbf{w}_1, \mathbf{w}_2 \in \mathbb{R}^d$ as follows:

$$a_{1:n} := \exp(\mathbf{Q}\mathbf{w}_1) \in (0, \infty)^n, \quad t_{1:n} := \text{sigmoid}(\mathbf{Q}\mathbf{w}_2) \in (0, 1)^n. \quad (10)$$

We use $\exp(\cdot)$ and $\text{sigmoid}(\cdot)$ to satisfy the mathematical domains of $a_i \in (0, \infty)$ and $t_i \in (0, 1)$, respectively. The $\text{sigmoid}(\cdot)$ for $t_i$ is selected as it also performed best empirically.

## 3.2 MODEL TRAINING

**Notation.** Without loss of generality, we assume that the observed edges are indexed before unobserved ones. Based on this, we denote each partition of the correlation matrix $\mathbf{R}$ as follows:

$$\mathbf{R} = \begin{bmatrix} \mathbf{R}_{00} & \mathbf{R}_{01} \\ \mathbf{R}_{10} & \mathbf{R}_{11} \end{bmatrix}, \quad \text{such that } \mathbf{R}_{00} \in [-1,1]^{m \times m} \text{ and } \mathbf{R}_{11} \in [-1,1]^{(n-m) \times (n-m)}, \quad (11)$$

where the subscripts 0 and 1 denote the sets of observed $\mathcal{E}_{\text{obs}}$ and unobserved edges $\mathcal{E}_{\text{miss}}$, respectively. Thus, the upper-left submatrix $\mathbf{R}_{00}$ corresponds to the training set, where the true signs are known. Also, in order to match the domain between our marginal distributions defined on the interval $(0,1)$ and the observed labels in $\{-1, +1\}$, we remap the negative labels $(-1)$ to zeros.

**Label Smoothing.** Eqs. (6) and (7) illustrate the marginal PDF is undefined at the boundary points 0 and 1, and the CDF evaluates to constants ($F_i(0; a_i, t_i) = 0$ and $F_i(1; a_i, t_i) = 1$), regardless of its parameters $a_i$ and $t_i$. Consequently, this provides no informative gradient for training the parameters of the marginal distributions. To resolve these issues and enable effective training, we apply *the label smoothing* (Müller et al., 2019). We map the original hard labels in $\{-1, +1\}$ to the smooth labels, denoted by $\bar{y}_i$:

$$\bar{y}_i := \begin{cases} \eta & \text{if } y_i = -1 \\ 1 - \eta & \text{if } y_i = +1 \end{cases}, \quad (12)$$

where $\eta \in (0, 0.5)$ is a hyperparameter. By preventing evaluation at the boundaries, this method guarantees an informative gradient for the marginal parameters. Similarly, we denote the smooth uniform variable, which serves as the input to the Gaussian copula, as $\bar{u}_i = F_i(\bar{y}_i; a_i, t_i)$.

**Loss Function.** We train our model $\mathcal{H}$ via maximum likelihood estimation by minimizing the following negative log-likelihood loss, formulated with the smooth labels $\bar{y}_i$:

$$\mathcal{L} := -\log \mathcal{H}'(\bar{y}_{1:m}; a_{1:m}, t_{1:m}, \mathbf{R}_{00}) = -\log \left( c(\bar{u}_{1:m}; \mathbf{R}_{00}) \prod_{i=1}^{m} f_i(\bar{y}_i; a_i, t_i) \right)$$

$$= \frac{1}{2} \log \det \mathbf{R}_{00} + \frac{1}{2} \mathbf{z}_{\text{obs}}^\top (\mathbf{R}_{00}^{-1} - \mathbf{I}_m) \mathbf{z}_{\text{obs}} - \sum_{i=1}^{m} \log f_i(\bar{y}_i; a_i, t_i), \quad (13)$$

where $\mathbf{z}_{\text{obs}} := (\Phi^{-1}(\bar{u}_1), \Phi^{-1}(\bar{u}_2), \cdots, \Phi^{-1}(\bar{u}_m)) \in \mathbb{R}^m$, and $\mathbf{R}_{00} \in [-1,1]^{m \times m}$ is the observed portion of the correlation matrix $\mathbf{R}$ as in Eq. (11).

**Generalization to Unobserved Edges.** Although we train our model using only the observed edges in the graph, the signed graph encoder (implemented as SNEA (Li et al., 2020) in Fig. 1) generalizes this learned knowledge to all nodes. Crucially, we adhere to the standard setting of link sign prediction. Under this standard setting, the objective is to determine the sign of a relationship given the premise that the edge structurally exists. Therefore, an *unobserved* edge under our context refers to an existing edge with its sign unknown, distinct from the regular link prediction task. This formulation allows us to directly compute not only their edge embeddings but also their marginal distributions via the learned linear projections. Consequently, our model $\mathcal{H}$ is applicable to the entire $\mathbf{R}$ as in Eq. (9), not just to its observed portion $\mathbf{R}_{00}$.

## 3.3 EFFICIENT INFERENCE

**Direct Inference.** To predict the sign of an unseen edge, we may perform inference by directly sampling from the fitted conditional distribution, given the known labels in $\mathcal{E}_{\text{obs}}$. Relying on a fundamental property that a conditional distribution of a Gaussian given another Gaussian is also a Gaussian, we may perform a naive sampling analytically from our Gaussian copula. Specifically, for $\mathbf{z}_{\text{miss}} := (\Phi^{-1}(u_{m+1}), \Phi^{-1}(u_{m+2}), \cdots, \Phi^{-1}(u_n))$, the conditional Gaussian distribution from which we sample is given by:

$$\mathbf{z}_{\text{miss}} | \mathbf{z}_{\text{obs}} \sim \mathcal{N}(\mathbf{R}_{10} \mathbf{R}_{00}^{-1} \mathbf{z}_{\text{obs}}, \mathbf{R}_{11} - \mathbf{R}_{10} \mathbf{R}_{00}^{-1} \mathbf{R}_{01}), \quad (14)$$

following the CopulaGNN (Ma et al., 2021).

To be more specific, to perform inference on an unobserved edge $e_j$ for $j \in \{m+1, ..., n\}$, we first draw an element of sample, $z_j \in \mathbb{R}$ (corresponding to edge $e_j$), from the conditional distribution in

Eq. (14). This element is then converted into a probabilistic score within the domain of marginal distribution $F_j(\cdot; a_j, t_j)$, which indicates the probability of a positive edge sign. The predicted label $\hat{y}_j$ and the probabilistic score $s_j$ are jointly expressed as:

$$\hat{y}_j := \begin{cases} +1 & \text{if } s_j \geq 0.5 \\ -1 & \text{if } s_j < 0.5 \end{cases}, \quad \text{such that} \quad s_j := F_j^{-1}(\Phi(z_j); a_j, t_j). \tag{15}$$

However, this direct sampling from the conditional distribution is prohibitively inefficient, mainly due to the high computational and memory cost of inverting $\mathbf{R}_{00} \in [-1, 1]^{m \times m}$. In practice, naively performing this inference step on a large dataset often leads to out-of-memory.

**Woodbury Reformulation for Efficient Inference.** To address this inefficiency, we start with the following decomposition for the correlation matrix $\mathbf{R}$, which is enabled by the property that the normalization function $\nu(\cdot)$ is a congruence transformation (Eq. (8)):

$$\mathbf{R} = \mathbf{D}^{-1}\mathbf{\Sigma}\mathbf{D}^{-1} = \mathbf{P}\mathbf{P}^\top + \mathbf{K}, \quad \text{such that} \quad \mathbf{P} := \mathbf{D}^{-1}\mathbf{Q} \quad \text{and} \quad \mathbf{K} := \epsilon\mathbf{D}^{-2}. \tag{16}$$

Then, we apply the following *Woodbury matrix identity* to $\mathbf{R}^{-1}$ to derive the equivalent result, not by inverting an huge $m \times m$ matrix, but a much smaller one.

**Theorem 3.1** (Woodbury Matrix Identity (Woodbury, 1950)). *Provided the matrices are conformable, the following equation holds:*

$$(\mathbf{A} + \mathbf{U}\mathbf{C}\mathbf{V})^{-1} = \mathbf{A}^{-1} - \mathbf{A}^{-1}\mathbf{U}(\mathbf{C}^{-1} + \mathbf{V}\mathbf{A}^{-1}\mathbf{U})^{-1}\mathbf{V}\mathbf{A}^{-1}. \tag{17}$$

Applying the Woodbury matrix identity, the inverse of correlation is reformulated as:

$$\mathbf{R}^{-1} = (\mathbf{K} + \mathbf{P}\mathbf{P}^\top)^{-1} = \mathbf{K}^{-1} - \mathbf{K}^{-1}\mathbf{P}\mathbf{S}^{-1}\mathbf{P}^\top\mathbf{K}^{-1}, \quad \text{such that } \mathbf{S} := \mathbf{I}_d + \mathbf{P}^\top\mathbf{K}^{-1}\mathbf{P}. \tag{18}$$

Therefore, the computational bottleneck is simplified to finding the inverse of a significantly smaller $d \times d$ matrix $\mathbf{S}$. Note that the positive definiteness of $\mathbf{R}$ guarantees the existence of $\mathbf{S}^{-1}$ by *the matrix determinant lemma* (Ding & Zhou, 2007). This makes our approach particularly desirable, since the computational overhead relies only on the edge embedding size $d$, which is a controllable hyperparameter, instead of the graph size $n$.

With a slight abuse of notation, the correlation matrix $\mathbf{R}$ is constructed as:

$$\mathbf{R} = \begin{bmatrix} \mathbf{R}_{00} & \mathbf{R}_{01} \\ \mathbf{R}_{10} & \mathbf{R}_{11} \end{bmatrix} = \begin{bmatrix} \mathbf{P}_0\mathbf{P}_0^\top + \mathbf{K}_0 & \mathbf{P}_0\mathbf{P}_1^\top \\ \mathbf{P}_1\mathbf{P}_0^\top & \mathbf{P}_1\mathbf{P}_1^\top + \mathbf{K}_1 \end{bmatrix}, \tag{19}$$

allowing the conditional distribution to be efficiently derived as a low-rank multivariate Gaussian. We term this method *the Woodbury Reformulation*.

**Theorem 3.2** (Woodbury Reformulation). *To improve computational efficiency, Eq. (14) can be reformulated to the low-rank multivariate Gaussian distribution as follows:*

$$\mathbf{z}_{\text{miss}}|\mathbf{z}_{\text{obs}} \sim \mathcal{N}(\mathbf{P}_1\mathbf{S}_0^{-1}\mathbf{P}_0^\top\mathbf{K}_0^{-1}\mathbf{z}_{\text{obs}}, \mathbf{P}_1\mathbf{S}_0^{-1}\mathbf{P}_1^\top + \mathbf{K}_1). \tag{20}$$

*Proof.* See Appendix D. □

## 4 CONVERGENCE ANALYSIS

In this section, we theoretically analyze the convergence of our proposed method. We show that our approach achieves *a linear convergence*, which means the error in successive steps of the iterative optimization process decreases by a constant factor. Starting with the formal definition of linear convergence, we prove that our method satisfies this property. Its practical impact on the fast convergence is empirically demonstrated in Sec. 5.

**Definition 4.1** (Linear Convergence). A function $f : \Omega \to \mathbb{R}$ is said to be linearly convergent if there exists a convergence rate $r \in (0, 1)$, for each optimization step $k \in \mathbb{N}$, such that:

$$f(x_k) - f^* \leq r(f(x_{k-1}) - f^*), \quad \text{where } f^* := \min_{x \in \Omega} f(x). \tag{21}$$

If we consider our loss function $\mathcal{L}$ (Eq. (13)) as $f$ above, and iterate Eq. (21) with the gradient descent optimization steps $k = 1, \ldots, K$, then we obtain the following inequality:

$$\mathcal{L}_K - \mathcal{L}^* \leq r^K(\mathcal{L}_0 - \mathcal{L}^*), \tag{22}$$

where $\mathcal{L}_k$ indicates the loss value after the step $k$ and $\mathcal{L}^*$ is the lowest possible loss. It is important to note that term $\mathcal{L}_0 - \mathcal{L}^*$ can be interpreted as the initial model error before training, while $\mathcal{L}_K - \mathcal{L}^*$ represents the model error achieved after $K$ epochs of training. Therefore, if the loss function achieves linear convergence, it implies that the error of model decreases at an exponentially fast rate ($r^K$) throughout the entire training process. Here, a smaller value of $r$ indicates faster convergence. Consequently, we can verify our hypothesis—that directly modeling the inter-edge correlation leads to an accelerated convergence rate—by proving that our loss is linearly convergent.

To prove the linear convergence of our loss function $\mathcal{L}$, we introduce the following theorem.

**Theorem 4.2** (Linear Convergence under the Gradient Descent (Karimi et al., 2016))**.** *A function satisfying both $L$-smoothness and the $\mu$-PL (Polyak-Lojasiewicz) condition converges linearly with a convergence rate $r = 1 - \mu/L$ under the gradient descent optimization.*

We theoretically justify the linear convergence of our model by demonstrating that its loss function satisfies both $L$-smoothness and $\mu$-PL (Polyak, 1963; Lojasiewicz, 1963) condition, with respect to the correlation $\mathbf{R}$. Here, $L$-smoothness signifies that the gradient is $L$-Lipschitz continuous, and $\mu$-PL condition is known as a weaker version of strong convexity. It is worth to note that the Gramian structure (Eq. (8)) and label smoothing (Eq. (12)) we introduce are essential to Theorem 4.2, which in turn establishes the theoretical soundness of our proposed method. The formal definitions of $L$-smoothness, $\mu$-PL condition, and the proof of Theorem 4.2 are provided in Appendix E.

## 5 EXPERIMENTS

To strictly validate the effectiveness of our proposed method, we conduct a comprehensive evaluation covering performance and scalability analysis, ablation studies to verify the contribution of each model component, and hyperparameter sensitivity analysis.

### 5.1 EXPERIMENTAL SETUP

We compare our method with state-of-the-art baselines, including GCN (Kipf & Welling, 2017), SGCN (Derr et al., 2018), SNEA (Li et al., 2020), SDGNN (Huang et al., 2021), TrustSGCN (Kim et al., 2023), SLGNN (Li et al., 2023), SE-SGformer (Li et al., 2025), and SGAAE (Nakis et al., 2025), on multiple real-world datasets including BitcoinAlpha (Kumar et al., 2016), BitcoinOTC (Kumar et al., 2018), WikiElec (Leskovec et al., 2010b), WikiRfa (West et al., 2014), SlashDot, and Epinions (Leskovec et al., 2010a). We report AUC and (Macro) F1 as our primary metrics, consistent with previous studies. All results are averaged over 10 distinct data splits to ensure statistical reliability. Further experimental details and supplementary results are provides in Appendix F.

**Backbone Encoder.** Our proposed method is agnostic to the choice of backbone SGNN encoder. To achieve superior scalability in terms of both time and memory compared to alternatives, we select SNEA (Li et al., 2020) as our primary backbone encoder. SNEA is a graph attention framework grounded in balance theory that learns two distinct representations for each node: a balanced embedding and an unbalanced embedding. These embeddings are updated hierarchically using a signed graph attentional layer. Specifically, a node's balanced embedding is updated by aggregating the balanced embeddings from its positive neighbors and the unbalanced embeddings from its negative neighbors. Conversely, the unbalanced embedding is updated by aggregating unbalanced embeddings from positive neighbors and balanced embeddings from negative neighbors. Finally, SNEA concatenates these two learned representations to produce the final node embedding.

### 5.2 PERFORMANCE AND SCALABILITY

In this section, we evaluate the scalability and link sign prediction performance of our method. We first verify the computational efficiency and convergence speed specifically against the SNEA backbone. Then, we demonstrate the method's efficiency and competitive performance through a comprehensive comparison with state-of-the-art baselines.

Table 1: Performance and scalability comparison: SNEA (backbone) vs. CopulaLSP (ours).

| | | BitcoinAlpha | | BitcoinOTC | | WikiElec | | WikiRfa | | SlashDot | | Epinions | |
|---|---|---|---|---|---|---|---|---|---|---|---|---|---|
| | | SNEA | Ours | SNEA | Ours | SNEA | Ours | SNEA | Ours | SNEA | Ours | SNEA | Ours |
| Training | Epoch to converge | 325.5 | 56.7 | 284.4 | 65.1 | 511.2 | 201.6 | 505.8 | 164.1 | 519.9 | 41.7 | 514.8 | 59.7 |
| | Time per epoch (sec) | 0.40 | 0.05 | 0.61 | 0.05 | 2.57 | 0.08 | 4.29 | 0.11 | 8.99 | 0.29 | 12.74 | 0.41 |
| | Total time (sec) | 131.2 | 3.0 | 174.9 | 3.6 | 1316.1 | 16.2 | 2174.1 | 18.0 | 4676.5 | 12.4 | 6574.4 | 24.7 |
| | Time speedup | - | 44× | - | 48× | - | 81× | - | 121× | - | 379× | - | 266× |
| | GPU memory (GB) | 0.60 | 0.74 | 0.68 | 0.82 | 1.34 | 1.47 | 1.90 | 2.08 | 4.96 | 5.08 | 7.03 | 7.20 |
| Inference | AUC | 0.866 | 0.864 | 0.886 | 0.885 | 0.872 | 0.877 | 0.859 | 0.862 | 0.885 | 0.884 | 0.898 | 0.899 |
| | Macro F1 | 0.669 | 0.716 | 0.755 | 0.771 | 0.754 | 0.768 | 0.738 | 0.751 | 0.770 | 0.771 | 0.809 | 0.810 |
| | Total time (sec) | 1.12 | 0.07 | 1.54 | 0.07 | 2.22 | 0.09 | 2.87 | 0.10 | 5.35 | 0.23 | 6.98 | 0.31 |
| | Time speedup | - | 16× | - | 22× | - | 25× | - | 29× | - | 23× | - | 23× |
| | GPU memory (GB) | 0.55 | 0.69 | 0.62 | 0.76 | 1.20 | 1.33 | 1.68 | 1.85 | 4.28 | 4.40 | 6.03 | 6.26 |

Table 2: Overall link sign prediction performance. OOM indicates out-of-memory.

| | BitcoinAlpha | | BitcoinOTC | | WikiElec | | WikiRfa | | SlashDot | | Epinions | |
|---|---|---|---|---|---|---|---|---|---|---|---|---|
| | AUC | F1 | AUC | F1 | AUC | F1 | AUC | F1 | AUC | F1 | AUC | F1 |
| GCN | 0.755 | 0.562 | 0.789 | 0.636 | 0.746 | 0.597 | 0.720 | 0.569 | 0.647 | 0.505 | 0.760 | 0.614 |
| SGCN | 0.846 | 0.673 | 0.880 | 0.737 | 0.876 | 0.749 | 0.831 | 0.712 | 0.737 | 0.624 | 0.880 | 0.752 |
| SNEA | 0.866 | 0.669 | 0.886 | 0.755 | 0.872 | 0.754 | 0.859 | 0.738 | 0.885 | 0.770 | 0.898 | 0.809 |
| SDGNN | 0.831 | 0.674 | 0.869 | 0.750 | 0.874 | 0.751 | 0.856 | 0.731 | OOM | OOM | OOM | OOM |
| TrustSGCN | 0.871 | 0.667 | 0.887 | 0.750 | 0.858 | 0.724 | 0.844 | 0.709 | OOM | OOM | OOM | OOM |
| SLGNN | 0.864 | 0.716 | 0.903 | 0.818 | 0.887 | 0.770 | 0.874 | 0.748 | 0.890 | 0.771 | OOM | OOM |
| SGAAE | 0.839 | 0.683 | 0.875 | 0.759 | 0.873 | 0.754 | 0.851 | 0.724 | OOM | OOM | OOM | OOM |
| SE-SGformer | 0.848 | 0.699 | 0.866 | 0.756 | 0.832 | 0.714 | 0.794 | 0.678 | OOM | OOM | OOM | OOM |
| CopulaLSP (ours) | 0.864 | 0.716 | 0.885 | 0.771 | 0.877 | 0.768 | 0.862 | 0.751 | 0.884 | 0.771 | 0.899 | 0.810 |

Table 3: Overall time and memory scalability comparison on WikiElec and WikiRfa.

| | WikiElec | | | WikiRfa | | |
|---|---|---|---|---|---|---|
| | Training (sec) | Inference (sec) | GPU (GB) | Training (sec) | Inference (sec) | GPU (GB) |
| GCN | 22.4 | 2.80 | 1.53 | 27.7 | 3.62 | 2.23 |
| SGCN | 2261.8 | 0.08 | 1.46 | 4718.8 | 0.11 | 2.29 |
| SNEA | 1316.1 | 2.22 | 1.34 | 2174.1 | 2.87 | 1.90 |
| SDGNN | 66.2 | 5.73 | 2.20 | 92.5 | 7.17 | 3.79 |
| TrustSGCN | 640.6 | 29.34 | 1.71 | 1561.0 | 46.60 | 1.90 |
| SLGNN | 59.0 | 0.06 | 1.83 | 87.0 | 0.09 | 3.00 |
| SGAAE | 40.1 | 0.57 | 2.09 | 37.6 | 0.97 | 3.27 |
| SE-SGformer | 1136.6 | 0.62 | 6.32 | 1610.0 | 0.78 | 15.42 |
| CopulaLSP (ours) | 16.2 | 0.09 | 1.47 | 18.0 | 0.10 | 2.08 |

**Comparative Analysis with SNEA.** As reported in Tab. 1, our method converges significantly faster than SNEA at training. This highlights the exceptional time scalability of our method, which is a critical advantage for real-world applications that involve large-scale graphs. Our method is also significantly faster than baselines at inference. SNEA relies on a two-stage procedure, learning node embeddings and then training a separate classifier on them, while our method simplifies this process into a single step of computing and sampling from a conditional probability distribution. Regarding memory consumption, our method requires slightly (about 200MB) more GPU memory than SNEA. However, this additional cost is primarily attributable to the linear projection (Eq. (10)) with a fixed number of parameters, not scaling with the graph size. To sum up, our method offers a decent trade-off, achieving significantly faster training and inference, paying this modest additional memory cost. Moreover, the performance gain relative to the SNEA backbone provides compelling evidence that our correlation modeling yields substantial practical benefits beyond mere computational efficiency.

**Overall Performance and Scalability.** We evaluate the overall link sign prediction performance and scalability of our method against a broader range of baselines, including the state-of-the-art model SLGNN. As shown in Tab. 2, our model achieves AUC and F1 scores that are competitive with most baselines. This is particularly significant when interpreted in the context of scalability on large-scale graphs. Most recent models encounter out-of-memory (OOM) issues and fail to operate on large datasets like SlashDot and Epinions. In contrast, our method demonstrates remarkable efficiency, as evidenced by the results on the Wiki dataset in Tab. 3. This highlights that our method not only maintains competitive performance but also possesses the robust scalability required for real-world applications, an advantage that remains consistent across other datasets (see Appendix F.1 for additional experimental results).

Table 4: Ablation on the correlation matrix: Identity (no correlation) vs. Gramian (ours)

| | BitcoinAlpha | | BitcoinOTC | | WikiElec | | WikiRfa | | SlashDot | | Epinions | |
|---|---|---|---|---|---|---|---|---|---|---|---|---|
| Correlation | Identity | Gram | Identity | Gram | Identity | Gram | Identity | Gram | Identity | Gram | Identity | Gram |
| AUC | 0.830 | 0.865 | 0.886 | 0.885 | 0.876 | 0.877 | 0.861 | 0.862 | 0.885 | 0.884 | 0.900 | 0.899 |
| F1 | 0.665 | 0.713 | 0.747 | 0.770 | 0.737 | 0.766 | 0.720 | 0.749 | 0.735 | 0.768 | 0.782 | 0.810 |
| Epoch to converge | 158.7 | 55.8 | 202.2 | 61.2 | 136.5 | 204.6 | 119.7 | 135.6 | 105.3 | 70.8 | 127.2 | 58.2 |
| Training time (sec) | 8.21 | 2.97 | 10.98 | 3.41 | 11.75 | 16.51 | 15.72 | 15.25 | 35.35 | 21.87 | 58.67 | 24.91 |

Table 5: Ablation on the Woodbury reformulation. OOM indicates out-of-memory.

| | BitcoinAlpha | | BitcoinOTC | | WikiElec | | WikiRfa | | SlashDot | | Epinions | |
|---|---|---|---|---|---|---|---|---|---|---|---|---|
| Woodbury refomulation | w/o | w/ | w/o | w/ | w/o | w/ | w/o | w/ | w/o | w/ | w/o | w/ |
| AUC | 0.864 | 0.865 | 0.884 | 0.885 | OOM | 0.877 | OOM | 0.862 | OOM | 0.884 | OOM | 0.899 |
| F1 | 0.712 | 0.713 | 0.772 | 0.770 | OOM | 0.766 | OOM | 0.749 | OOM | 0.768 | OOM | 0.810 |
| Inference time (sec) | 0.26 | 0.07 | 0.64 | 0.07 | OOM | 0.09 | OOM | 0.11 | OOM | 0.24 | OOM | 0.32 |
| Inference speedup | - | 4× | - | 9× | - | - | - | - | - | - | - | - |
| Inference GPU usage (GB) | 3.12 | 0.62 | 6.47 | 0.69 | 24.16 | 1.26 | 70.18 | 1.77 | 597.19 | 4.33 | 1182.54 | 6.18 |
| Inference GPU gain | - | 5× | - | 9× | - | 19× | - | 40× | - | 138 × | - | 191× |

## 5.3 ABLATION STUDY

We conduct an ablation study to validate the effectiveness of our model's key components: the Gramian-based correlation matrix (Eq. (8)) and the Woodbury reformulation (Eq. (20)).

**Gramian-based Correlation.** We compare the performance of using our Gramian-based correlation against fixing it as an identity matrix (*i.e.,* no inter-edge correlation). As shown in Tab. 4, learning the correlation matrix indeed improves the classification performance (F1). Furthermore, it also accelerates the convergence speed (Epoch to converge) in most cases, with the exception of the Wiki dataset. These empirical findings align well with our theoretical convergence analysis in Sec. 4.

**Woodbury Reformulation.** We evaluate the effect of the Woodbury reformulation, which was introduced to reduce the computational cost of matrix inversion. Tab. 5 confirms that this reformulation leads to a significant reduction in time and memory costs at inference, with no degradation in model performance.

## 5.4 HYPERPARAMETER STUDY

We primarily analyze the impact of the label smoothing $\eta$ on convergence speed and performance. See Appendix F.1 for more results on other hyperparameters ($\epsilon$ and embedding size $d$).

**Convergence Study.** We examine how $\eta$ affects model convergence speed. Fig. 3 demonstrates that increasing $\eta$ accelerates the convergence speed. This empirical observation is supported by our theoretical analysis; as derived in Appendix E, a larger $\eta$ constrains the norm of the latent variables, which consequently lowers the upper bound of the matrix norm $\|\mathbf{z}_{\mathrm{obs}}\mathbf{z}_{\mathrm{obs}}^{\top}\|$. This reduction leads to a smaller convergence rate $r$, thereby resulting in faster convergence.

**Performance Study.** We further evaluate the sensitivity of the model performance with respect to $\eta$. Fig. 4 demonstrates that the model maintains robust performance across a wide range of $\eta$ values.

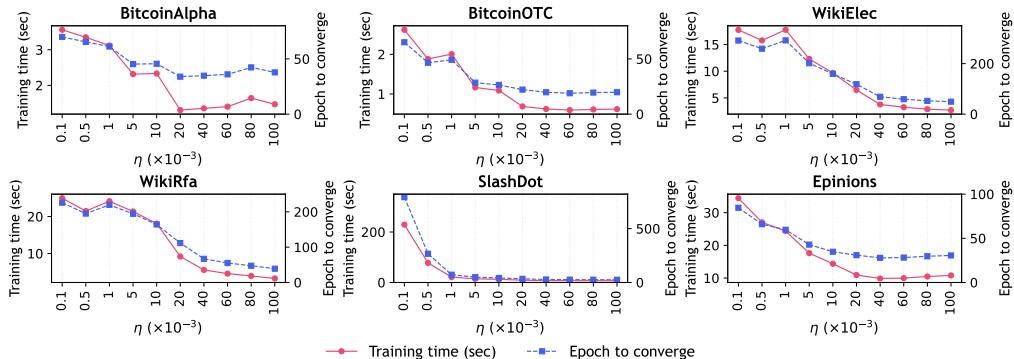

Figure 3: Convergence of CopulaLSP varying hyperparameter $\eta$.

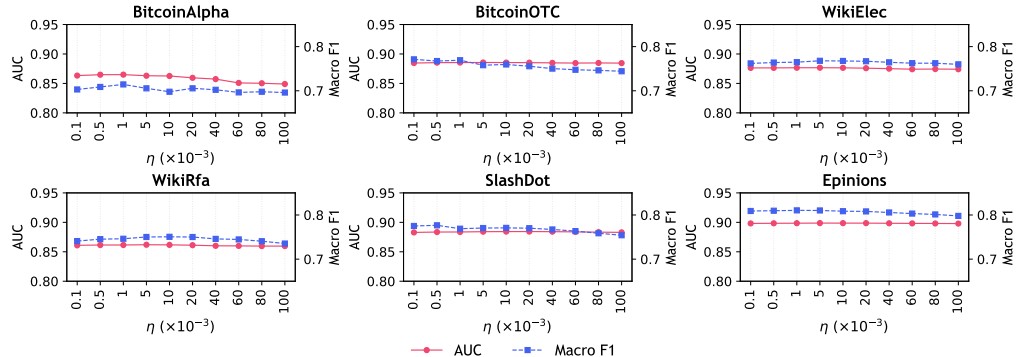

Figure 4: Performance of CopulaLSP varying hyperparameter $\eta$.

# 6 RELATED WORK

**Signed Graph Neural Networks.** A key challenge in signed graph learning is that negative edges violate the graph homophily assumption. Early work, such as SGCN (Derr et al., 2018) and SDGNN (Huang et al., 2021), addressed this by incorporating balance theory (Heider, 1946) or status theory (Easley & Kleinberg, 2010). Subsequently, SiGAT (Huang et al., 2019) and SNEA (Li et al., 2020) further enhanced representation quality by applying the graph attention (Velickovic et al., 2018). MSGNN (He et al., 2022) and SLGNN (Li et al., 2023) refines representation by emphasizing low-frequency signals through the utilization of graph spectral filtering (Defferrard et al., 2016; Kim et al., 2025a). More recent efforts, such as TrustSGCN (Kim et al., 2023), SE-SGformer (Li et al., 2025), and SGAAE (Nakis et al., 2025), have shifted focus toward reliability and explainability, using trust-aware propagation and explainable architectures. Despite these advances, a common limitation across these methods is their separate handling of positive and negative edges during aggregation, which overlooks the statistical dependencies that may exist between them.

**Copula in Deep Learning.** A key advantage of copula functions (Sklar, 1959; Tagasovska et al., 2019) is their ability to separate a joint probability distribution into two distinct components: the univariate marginal distributions and the dependency structure that couples them. This decomposition is particularly powerful for modeling complex data distributions, applied to diverse tasks such as density estimation (Ling et al., 2020; Huk et al., 2025), representation learning (Wieczorek et al., 2018; Lu & Peltonen, 2022), generative modeling (Ng et al., 2021; Liu et al., 2025), and forecasting on time-series (Drouin et al., 2022; Ashok et al., 2024). For graph representation learning, CopulaGNN (Ma et al., 2021) leveraged a Gaussian copula to explicitly model the correlation structure among nodes. Their demonstrated performance gain provides a strong motivation for our work.

# 7 CONCLUSION

In this work, we present CopulaLSP, a novel and scalable framework that models the inter-edge correlation for link sign prediction. Our approach takes the Gaussian copula to couple marginal relaxed Bernoulli distributions with the edge correlation derived from the Gramian of edge embeddings. Furthermore, we introduce the Woodbury reformulation to maximize the time and memory efficiency of the sampling-based inference. We not just theoretically verify fast convergence of the proposed method, but also empirically demonstrate its competitive performance and memory efficiency on real-world data. We believe this work opens promising avenues for applying efficient inter-edge correlation modeling to a wider range of graph-based tasks.

**Limitations.** We have assumed a static graph structure throughout this work, as it is inherent to semi-supervised learning on graphs. Applying our approach to dynamic and bipartite graphs would be a promising future direction, *e.g.*, for social recommendation systems.

ACKNOWLEDGMENTS

This work was supported by SOFT Foundry Institute at SNU, Youlchon Foundation, National Research Foundation of Korea (NRF) grants (RS-2021-NR05515, RS-2024-00336576, RS-2023-0022663, RS-2025-25399604) and Institute for Information & communication Technology Planning & evaluation (IITP) grants (RS-2022-II220264, RS-2024-00353131) by the government of Korea.

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

APPENDIX

## A  RELAXED BERNOULLI DISTRIBUTION

Note that simplified PDF of the relaxed Bernoulli distribution can be written as:

$$f_{\text{simple}}(x; a, t) := \frac{atx^{t-1}(1-x)^{t-1}}{(a(1-x)^t + x^t)^2},\tag{23}$$

and remark that it is well-defined on $(0, 1)$.

Now, we can show that CDF of relaxed Bernoulli distribution $F(\cdot; a, t)$ is well-defined by proving $dF/dx = f_{\text{simple}}(x; a, t)$. For the sake of simplicity, we introduce $Q_1 := x^t$, $Q_2 := a(1 - x)^t$, $Q_3 := Q_1 + Q_2$, and $Q_i' := dQ_i/dx$.

From the quotient rule, we obtain:

$$\frac{dF}{dx} = \frac{d}{dx}\frac{Q_1}{Q_3} = \frac{Q_1'Q_3 - Q_1Q_3'}{(Q_3)^2}.\tag{24}$$

A straightforward calculation shows that:

$$Q_1'Q_3 - Q_1Q_3' = tx^{t-1}[(x^t + a(1-x)^t) - (x^t - ax(1-x)^{t-1})]\tag{25}$$

$$= tx^{t-1}[a(1-x)^t + ax(1-x)^{t-1}]\tag{26}$$

$$= tx^{t-1} \cdot a(1-x)^{t-1}[(1-x) + x]\tag{27}$$

$$= atx^{t-1}(1-x)^{t-1}.\tag{28}$$

Therefore,

$$\frac{dF}{dx} = \frac{atx^{t-1}(1-x)^{t-1}}{(a(1-x)^t + x^t)^2} = f_{\text{simple}}(x; a, t).\tag{29}$$

The derivation of the ICDF from the CDF is considerably more intuitive. Consider the following:

$$F(y; a, t) = x = \frac{y^t}{a(1-y)^t + y^t} = \frac{1}{a(1/y - 1)^t + 1} \qquad \text{if } y \neq 0\tag{30}$$

$$\Rightarrow a\left(\frac{1}{y} - 1\right)^t = \frac{1}{x} - 1\tag{31}$$

$$\Rightarrow \left(\frac{1}{y} - 1\right)^t = a^{-1}\left(\frac{1}{x} - 1\right)\tag{32}$$

$$\Rightarrow \frac{1}{y} = 1 + a^{-1/t}\left(\frac{1}{x} - 1\right)^{1/t}\tag{33}$$

$$\Rightarrow y = \frac{1}{1 + a^{-1/t}\left(\frac{1}{x} - 1\right)^{1/t}} = \frac{x^{1/t}}{x^{1/t} + a^{-1/t}(1-x)^{1/t}} = F^{-1}(x; a, t).\tag{34}$$

## B  REPRESENTATIONAL POWER OF GRAMIAN-BASED CORRELATION

We experimentally demonstrate that our Gramian-based correlation design (Eq. (8)) can behave as *an ideal correlation matrix*. To validate our approach, we define an ideal correlation $\mathbf{R}^* \in \{-1, 0, 1\}^{n \times n}$ derived from the entire graph dataset as follows:

$$\mathbf{R}_{ij}^* := \begin{cases} 0 & \text{if } e_i, e_j \text{ have no shared node,} \\ +1 & \text{if } e_i, e_j \text{ have a shared node and } y_i = y_j, \\ -1 & \text{if } e_i, e_j \text{ have a shared node and } y_i \neq y_j, \end{cases}\tag{35}$$

where $y_i, y_j \in \{-1, +1\}$ are the true labels of the edges $e_i, e_j \in \mathcal{E}$, respectively.

However, this matrix is an ill-posed correlation matrix because it is not guaranteed to be positive definite. To address this, we leveraged the theory of *the nearest correlation matrix problem* (Higham,

2002; Borsdorf & Higham, 2010). We solve the following optimization problem using gradient descent to find *an approximated ideal factor* $\mathbf{Q} \in \mathbb{R}^{n \times d}$:

$$\mathbf{Q} := \underset{\mathbf{X} \in \mathbb{R}^{n \times d}}{\operatorname{argmin}} \|\nu\left(\mathbf{X}\mathbf{X}^\top + \epsilon\mathbf{I}\right) - \mathbf{R}^*\|^2. \tag{36}$$

Now we obtain *the approximated ideal correlation*, $\mathbf{R} = \nu(\mathbf{Q}\mathbf{Q}^\top + \epsilon\mathbf{I})$, based on our Gramian-based correlation design (Eq. (8)). Formally, $\mathbf{R}$ is the nearest (valid) correlation matrix of $\mathbf{R}^*$.

We then plug and fix this approximated ideal correlation into the Gaussian copula in our model to evaluate its performance. Notably, we observe near-perfect performances, achieving an AUC of 0.99 and a F1 of 0.95 on the BitcoinAlpha dataset.

This results show that the ideal form of correlation matrix is $\mathbf{R}^*$, and our Gramian-based correlation design has sufficient representational power to mimic the ideal correlation matrix $\mathbf{R}^*$.

## C    LOW-RANK GAUSSIAN DISTRIBUTION

*A low-rank Gaussian distribution* refers to a multivariate Gaussian distribution whose covariance matrix is expressed in the form $\mathbf{A}\mathbf{A}^\top + \mathbf{B}$, where $\mathbf{A}$ is a covariance factor (matrix) and $\mathbf{B}$ is a covariance diagonal (matrix). This low-rank covariance structure allows for the computationally efficient implementation of algorithms due to the Woodbury matrix identity. For details, refer to the `LowRankMultivariateNormal` class in the PyTorch library[1].

It is worthwhile to note that the Gramian-based correlation $\mathbf{R}$ can be derived as:

$$\mathbf{R} = \mathbf{D}^{-1}\mathbf{\Sigma}\mathbf{D}^{-1} = \mathbf{P}\mathbf{P}^\top + \mathbf{K} \quad \text{such that} \quad \mathbf{P} := \mathbf{D}^{-1}\mathbf{Q} \quad \text{and} \quad \mathbf{K} := \epsilon\mathbf{D}^{-2}. \tag{37}$$

Therefore, the Gaussian copula function of our method can be implemented in a computationally efficient manner.

It also holds true for inference. When considering the Woodbury reformulation in Eq. (20), the correlation term of the re-derived conditional probability distribution is equivalent to $\mathbf{P}_1\mathbf{S}_0^{-1}\mathbf{P}_1^\top + \mathbf{K}_1$. Since $\mathbf{S}_0^{-1}$ is positive definite and symmetric (thus allowing for Cholesky decomposition), $\mathbf{P}_1\mathbf{S}_0^{-1}\mathbf{P}_1^\top + \mathbf{K}_1$ can be re-derived into the following low-rank form:

$$\mathbf{P}_1\mathbf{S}_0^{-1}\mathbf{P}_1^\top + \mathbf{K}_1 = \mathbf{P}_1(\mathbf{L}\mathbf{L}^\top)\mathbf{P}_1^\top + \mathbf{K}_1 = (\mathbf{P}_1\mathbf{L})(\mathbf{P}_1\mathbf{L})^\top + \mathbf{K}_1. \tag{38}$$

Again, the conditional distribution can be implemented efficiently.

## D    WOODBURY REFORMULATION

**Theorem D.1** (Woodbury Reformulation). *To improve computational efficiency, Eq.* (14) *can be reformulated to low-rank multivariate Gaussian distribution as follows:*

$$\mathbf{z}_{\text{miss}}|\mathbf{z}_{\text{obs}} \sim \mathcal{N}(\mathbf{P}_1\mathbf{S}_0^{-1}\mathbf{P}_0^\top\mathbf{K}_0^{-1}\mathbf{z}_{\text{obs}}, \mathbf{P}_1\mathbf{S}_0^{-1}\mathbf{P}_1^\top + \mathbf{K}_1). \tag{39}$$

*Proof.* We aim to reformulate the mean $\mathbf{R}_{10}\mathbf{R}_{00}^{-1}\mathbf{z}_{\text{obs}}$ and covariance $\mathbf{R}_{11} - \mathbf{R}_{10}\mathbf{R}_{00}^{-1}\mathbf{R}_{01}$ of the conditional distribution $\mathbf{z}_{\text{miss}}|\mathbf{z}_{\text{obs}}$.

First, we apply the Woodbury matrix identity to $\mathbf{R}_{00}^{-1}$. Given the definition $\mathbf{R}_{00} = \mathbf{K}_0 + \mathbf{P}_0\mathbf{P}_0^\top$, its inverse is:

$$\mathbf{R}_{00}^{-1} = \mathbf{K}_0^{-1} - \mathbf{K}_0^{-1}\mathbf{P}_0(\mathbf{I}_d + \mathbf{P}_0^\top\mathbf{K}_0^{-1}\mathbf{P}_0)^{-1}\mathbf{P}_0^\top\mathbf{K}_0^{-1}. \tag{40}$$

Here, by defining $\mathbf{S}_0 := \mathbf{I}_d + \mathbf{P}_0^\top\mathbf{K}_0^{-1}\mathbf{P}_0$, the expression simplifies to:

$$\mathbf{R}_{00}^{-1} = \mathbf{K}_0^{-1} - \mathbf{K}_0^{-1}\mathbf{P}_0\mathbf{S}_0^{-1}\mathbf{P}_0^\top\mathbf{K}_0^{-1}. \tag{41}$$

The conditional mean is $\mathbf{R}_{10}\mathbf{R}_{00}^{-1}\mathbf{z}_{\text{obs}}$. We substitute $\mathbf{R}_{10} = \mathbf{P}_1\mathbf{P}_0^\top$ and the derived expression for $\mathbf{R}_{00}^{-1}$ into this equation.

$$\mathbf{R}_{10}\mathbf{R}_{00}^{-1}\mathbf{z}_{\text{obs}} = \left(\mathbf{P}_1\mathbf{P}_0^\top\right)\left(\mathbf{K}_0^{-1} - \mathbf{K}_0^{-1}\mathbf{P}_0\mathbf{S}_0^{-1}\mathbf{P}_0^\top\mathbf{K}_0^{-1}\right)\mathbf{z}_{\text{obs}} \tag{42}$$

$$= \left(\mathbf{P}_1\mathbf{P}_0^\top\mathbf{K}_0^{-1} - \mathbf{P}_1\mathbf{P}_0^\top\mathbf{K}_0^{-1}\mathbf{P}_0\mathbf{S}_0^{-1}\mathbf{P}_0^\top\mathbf{K}_0^{-1}\right)\mathbf{z}_{\text{obs}} \tag{43}$$

$$= \left(\mathbf{P}_1\mathbf{P}_0^\top\mathbf{K}_0^{-1} - \mathbf{P}_1(\mathbf{P}_0^\top\mathbf{K}_0^{-1}\mathbf{P}_0)\mathbf{S}_0^{-1}\mathbf{P}_0^\top\mathbf{K}_0^{-1}\right)\mathbf{z}_{\text{obs}}. \tag{44}$$

---

[1]`https://docs.pytorch.org/docs/stable/distributions.html`

From the definition of $\mathbf{S}_0$, we use the identity $\mathbf{P}_0^\top \mathbf{K}_0^{-1} \mathbf{P}_0 = \mathbf{S}_0 - \mathbf{I}_d$ and substitute it into the expression.

$$\mathbf{R}_{10}\mathbf{R}_{00}^{-1}\mathbf{z}_{\text{obs}} = \left(\mathbf{P}_1\mathbf{P}_0^\top\mathbf{K}_0^{-1} - \mathbf{P}_1(\mathbf{S}_0 - \mathbf{I}_d)\mathbf{S}_0^{-1}\mathbf{P}_0^\top\mathbf{K}_0^{-1}\right)\mathbf{z}_{\text{obs}} \tag{45}$$

$$= \left(\mathbf{P}_1\mathbf{P}_0^\top\mathbf{K}_0^{-1} - \mathbf{P}_1(\mathbf{S}_0\mathbf{S}_0^{-1} - \mathbf{S}_0^{-1})\mathbf{P}_0^\top\mathbf{K}_0^{-1}\right)\mathbf{z}_{\text{obs}} \tag{46}$$

$$= \left(\mathbf{P}_1\mathbf{P}_0^\top\mathbf{K}_0^{-1} - \mathbf{P}_1(\mathbf{I}_d - \mathbf{S}_0^{-1})\mathbf{P}_0^\top\mathbf{K}_0^{-1}\right)\mathbf{z}_{\text{obs}} \tag{47}$$

$$= \left(\mathbf{P}_1\mathbf{P}_0^\top\mathbf{K}_0^{-1} - (\mathbf{P}_1\mathbf{P}_0^\top\mathbf{K}_0^{-1} - \mathbf{P}_1\mathbf{S}_0^{-1}\mathbf{P}_0^\top\mathbf{K}_0^{-1})\right)\mathbf{z}_{\text{obs}} \tag{48}$$

$$= \mathbf{P}_1\mathbf{S}_0^{-1}\mathbf{P}_0^\top\mathbf{K}_0^{-1}\mathbf{z}_{\text{obs}}. \tag{49}$$

The conditional covariance is $\mathbf{R}_{11} - \mathbf{R}_{10}\mathbf{R}_{00}^{-1}\mathbf{R}_{01}$. We substitute $\mathbf{R}_{11} = \mathbf{P}_1\mathbf{P}_1^\top + \mathbf{K}_1$, $\mathbf{R}_{10} = \mathbf{P}_1\mathbf{P}_0^\top$, and $\mathbf{R}_{01} = \mathbf{P}_0\mathbf{P}_1^\top$.

$$\mathbf{R}_{11} - \mathbf{R}_{10}\mathbf{R}_{00}^{-1}\mathbf{R}_{01} = (\mathbf{P}_1\mathbf{P}_1^\top + \mathbf{K}_1) - (\mathbf{P}_1\mathbf{P}_0^\top)\mathbf{R}_{00}^{-1}(\mathbf{P}_0\mathbf{P}_1^\top). \tag{50}$$

From the derivation of the mean, we showed that $\mathbf{R}_{10}\mathbf{R}_{00}^{-1} = \mathbf{P}_1\mathbf{P}_0^\top\mathbf{R}_{00}^{-1} = \mathbf{P}_1\mathbf{S}_0^{-1}\mathbf{P}_0^\top\mathbf{K}_0^{-1}$. We substitute this result.

$$\mathbf{R}_{11} - \mathbf{R}_{10}\mathbf{R}_{00}^{-1}\mathbf{R}_{01} = (\mathbf{P}_1\mathbf{P}_1^\top + \mathbf{K}_1) - (\mathbf{P}_1\mathbf{S}_0^{-1}\mathbf{P}_0^\top\mathbf{K}_0^{-1})(\mathbf{P}_0\mathbf{P}_1^\top) \tag{51}$$

$$= \mathbf{P}_1\mathbf{P}_1^\top + \mathbf{K}_1 - \mathbf{P}_1\mathbf{S}_0^{-1}(\mathbf{P}_0^\top\mathbf{K}_0^{-1}\mathbf{P}_0)\mathbf{P}_1^\top. \tag{52}$$

Again, we use the identity $\mathbf{P}_0^\top\mathbf{K}_0^{-1}\mathbf{P}_0 = \mathbf{S}_0 - \mathbf{I}_d$.

$$\mathbf{R}_{11} - \mathbf{R}_{10}\mathbf{R}_{00}^{-1}\mathbf{R}_{01} = \mathbf{P}_1\mathbf{P}_1^\top + \mathbf{K}_1 - \mathbf{P}_1\mathbf{S}_0^{-1}(\mathbf{S}_0 - \mathbf{I}_d)\mathbf{P}_1^\top \tag{53}$$

$$= \mathbf{P}_1\mathbf{P}_1^\top + \mathbf{K}_1 - \mathbf{P}_1(\mathbf{S}_0^{-1}\mathbf{S}_0 - \mathbf{S}_0^{-1})\mathbf{P}_1^\top \tag{54}$$

$$= \mathbf{P}_1\mathbf{P}_1^\top + \mathbf{K}_1 - \mathbf{P}_1(\mathbf{I}_d - \mathbf{S}_0^{-1})\mathbf{P}_1^\top \tag{55}$$

$$= \mathbf{P}_1\mathbf{P}_1^\top + \mathbf{K}_1 - (\mathbf{P}_1\mathbf{P}_1^\top - \mathbf{P}_1\mathbf{S}_0^{-1}\mathbf{P}_1^\top) \tag{56}$$

$$= \mathbf{P}_1\mathbf{P}_1^\top + \mathbf{K}_1 - \mathbf{P}_1\mathbf{P}_1^\top + \mathbf{P}_1\mathbf{S}_0^{-1}\mathbf{P}_1^\top \tag{57}$$

$$= \mathbf{P}_1\mathbf{S}_0^{-1}\mathbf{P}_1^\top + \mathbf{K}_1. \tag{58}$$

By combining the derived mean and covariance, the conditional distribution is reformulated as:

$$\mathbf{z}_{\text{miss}}|\mathbf{z}_{\text{obs}} \sim \mathcal{N}(\mathbf{P}_1\mathbf{S}_0^{-1}\mathbf{P}_0^\top\mathbf{K}_0^{-1}\mathbf{z}_{\text{obs}}, \mathbf{P}_1\mathbf{S}_0^{-1}\mathbf{P}_1^\top + \mathbf{K}_1). \tag{59}$$

$\square$

# E  CONVERGENCE ANALYSIS

We introduce the formal definitions of *L-smoothness* and *$\mu$-PL (Polyak-Lojasiewicz) condition*.

**Definition E.1** (*L*-smoothness)**.** A differentiable function $f : \Omega \to \mathbb{R}$ is said to be *L*-smooth if its gradient $\nabla f$ is *L*-Lipschitz continuous. In other words, there exists a constant $L > 0$ such that:

$$\|\nabla f(x) - \nabla f(y)\| \leq L\|x - y\| \quad \forall x, y \in \Omega. \tag{60}$$

**Definition E.2** ($\mu$-PL Condition (Polyak, 1963; Lojasiewicz, 1963))**.** Suppose a differentiable function $f : \Omega \to \mathbb{R}$ has a minimum function value $f^*$. Then, we say $f$ satisfies the $\mu$-PL condition if the following holds for some $\mu > 0$:

$$\frac{1}{2}\|\nabla f(x)\|^2 \geq \mu(f(x) - f^*) \quad \forall x \in \Omega. \tag{61}$$

To examine the effect of correlation on model convergence, we begin our proof with the gradient of the loss $\mathcal{L}$ (Eq. (13)) with respect to $\mathbf{R}$:

$$\nabla_{\mathbf{R}}\mathcal{L} = \frac{1}{2}\mathbf{R}^{-1} - \frac{1}{2}\mathbf{R}^{-1}\mathbf{z}\mathbf{z}^\top\mathbf{R}^{-1}. \tag{62}$$

It should be noted that for the sake of simplicity, we have omitted the notation specifically introduced for the observed data.

**Lemma E.3.** $\nabla_{\mathbf{R}}\mathcal{L}$ *is Lipschitz continuous.*

*Proof.* Suppose that arbitrary two correlations $\mathbf{R}_1, \mathbf{R}_2 \in \mathcal{R}$ are given, where $\mathcal{R}$ is a set of our learnable correlations. Note that $\|\cdot\|_2$ is the spectral norm operator and $\|\cdot\|_F$ is the Frobenious norm operator. From the triangular inequality, we obtain:

$$2\|\nabla_{\mathbf{R}}\mathcal{L}(\mathbf{R}_1) - \nabla_{\mathbf{R}}\mathcal{L}(\mathbf{R}_2)\|_F \le \|\mathbf{R}_1^{-1} - \mathbf{R}_2^{-1}\|_F + \|\mathbf{R}_2^{-1}\mathbf{z}\mathbf{z}^\top\mathbf{R}_2^{-1} - \mathbf{R}_1^{-1}\mathbf{z}\mathbf{z}^\top\mathbf{R}_1^{-1}\|_F. \quad (63)$$

As our design of covariance matrix $\mathbf{\Sigma}$ (Eq. (8)) constrains the minimum eigenvalue to $\epsilon > 0$, and so any correlation matrix $\mathbf{R} \in \mathcal{R}$ also has a positive minimum eigenvalue by *Sylvester's law of inertia* (Sylvester, 1852), we can assume $\|\mathbf{R}^{-1}\|_2 \le \alpha$ for some constant $\alpha > 0$. Since $\mathbf{R}_1^{-1} - \mathbf{R}_2^{-1} = \mathbf{R}_1^{-1}(\mathbf{R}_2 - \mathbf{R}_1)\mathbf{R}_2^{-1}$, we can utilize a sub-multiplicativity of matrix norm as follows:

$$\|\mathbf{R}_1^{-1} - \mathbf{R}_2^{-1}\|_F \le \|\mathbf{R}_1^{-1}\|_2\|\mathbf{R}_2 - \mathbf{R}_1\|_F\|\mathbf{R}_2^{-1}\|_2 \le \alpha^2\|\mathbf{R}_1 - \mathbf{R}_2\|_F. \quad (64)$$

Similarly, we can also derive:

$$\mathbf{J} := \mathbf{R}_2^{-1}\mathbf{z}\mathbf{z}^\top\mathbf{R}_2^{-1} - \mathbf{R}_1^{-1}\mathbf{z}\mathbf{z}^\top\mathbf{R}_1^{-1} = \mathbf{R}_2^{-1}\mathbf{z}\mathbf{z}^\top(\mathbf{R}_2^{-1} - \mathbf{R}_1^{-1}) + (\mathbf{R}_2^{-1} - \mathbf{R}_1^{-1})\mathbf{z}\mathbf{z}^\top\mathbf{R}_1^{-1}. \quad (65)$$

Again, by sub-multiplicativity:

$$\|\mathbf{J}\|_F \le \|\mathbf{R}_2^{-1}\|_2\|\mathbf{z}\mathbf{z}^\top\|_F\|\mathbf{R}_2^{-1} - \mathbf{R}_1^{-1}\|_F + \|\mathbf{R}_2^{-1} - \mathbf{R}_1^{-1}\|_F\|\mathbf{z}\mathbf{z}^\top\|_F\|\mathbf{R}_1^{-1}\|_2. \quad (66)$$

Moreover, we can assume $\|\mathbf{z}\mathbf{z}^\top\| \le \beta$ for some $\beta > 0$. Consider the definition of $z_i$, *i.e.*, $z_i = \Phi^{-1}(u_i)$. Since $u_i$ is derived from a univariate Gaussian CDF $\Phi(\cdot)$ by the smooth label (Eq. (12)) which restricts the support of $\Phi(\cdot)$ to $[\eta, 1 - \eta]$, we can say $u_i$ is closed and bounded. In hence, $z_i = \Phi^{-1}(u_i)$ is also bounded and this implies $\|\mathbf{z}\mathbf{z}^\top\|_F$ is bounded. For instance, we can check $|\Phi^{-1}(1e-5)| < 4.3$, and it exemplifies assumption $\|\mathbf{z}\mathbf{z}^\top\|_F \le \beta$ is not strong. Hence, we have:

$$\|\mathbf{J}\|_F \le 2\alpha\beta\|\mathbf{R}_2^{-1} - \mathbf{R}_1^{-1}\|_F \le 2\alpha^3\beta\|\mathbf{R}_1 - \mathbf{R}_2\|_F. \quad (67)$$

Therefore, $\nabla_{\mathbf{R}}\mathcal{L}$ is Lipschitz continuous with respect to $\mathbf{R}$:

$$2\|\nabla_{\mathbf{R}}\mathcal{L}(\mathbf{R}_1) - \nabla_{\mathbf{R}}\mathcal{L}(\mathbf{R}_2)\|_F \le \|\mathbf{R}_1^{-1} - \mathbf{R}_2^{-1}\|_F + \|\mathbf{J}\|_F \quad (68)$$

$$\le \alpha^2\|\mathbf{R}_1 - \mathbf{R}_2\|_F + 2\alpha^3\beta\|\mathbf{R}_1 - \mathbf{R}_2\|_F \quad (69)$$

$$= (\alpha^2 + 2\alpha^3\beta)\|\mathbf{R}_1 - \mathbf{R}_2\|_F. \quad (70)$$

$\square$

**Lemma E.4.** $\mathcal{L}$ *satisfies the Polyak-Lojasiewicz condition with respect to* $\mathbf{R}$.

*Proof.* Redefine $\mathcal{L}$ with $\mathbf{A} := \mathbf{R}^{-1}$ as follows:

$$\tilde{\mathcal{L}}(\mathbf{A}) := \mathcal{L}(\mathbf{R}) = -\frac{1}{2}\log\det\mathbf{A} + \frac{1}{2}\mathbf{z}^\top(\mathbf{A} - \mathbf{I})\mathbf{z}. \quad (71)$$

Consider the gradient of $\tilde{\mathcal{L}}$ with respect to $\mathbf{A}$:

$$\nabla_{\mathbf{A}}\tilde{\mathcal{L}} = -\frac{1}{2}\mathbf{A}^{-1} + \frac{1}{2}\mathbf{z}\mathbf{z}^\top = -\mathbf{R}(\nabla_{\mathbf{R}}\mathcal{L})\mathbf{R}. \quad (72)$$

We can utilize a sub-multiplicativity of matrix norm as follows:

$$\|\nabla_{\mathbf{A}}\tilde{\mathcal{L}}\|_F = \|\mathbf{R}(\nabla_{\mathbf{R}}\mathcal{L})\mathbf{R}\|_F \le \|\mathbf{R}\|_2^2\|\nabla_{\mathbf{R}}\mathcal{L}\|_F = \lambda_{\max}(\mathbf{R})^2\|\nabla_{\mathbf{R}}\mathcal{L}\|_F \le m^2\|\nabla_{\mathbf{R}}\mathcal{L}\|_F, \quad (73)$$

where $\lambda_{\max}(\mathbf{R})$ is the maximum eigenvalue of correlation matrix $\mathbf{R}$, and remark that the maximum eigenvalues of correlation matrix has an upperbound, in fact, which is $m$. It is worth to noting that $\mathbf{z}^\top\mathbf{A}\mathbf{z}$ is trivially convex, and $-\log\det\mathbf{A}$ is *strongly convex* with respect to $\mathbf{A}$ since $\mathbf{R}$ has a positive minimum eigenvalue. So, $\tilde{\mathcal{L}}$ is also strongly convex. Karimi et al. (2016) shows a strong convexity implies a satisfaction of Polyak-Lojasiewicz condition. Hence, $\tilde{\mathcal{L}}$ satisfies the Polyak-Lojasiewicz condition for some constant $\tilde{\mu} > 0$:

$$\frac{1}{2}\|\nabla_{\mathbf{A}}\tilde{\mathcal{L}}\|_F^2 \ge \tilde{\mu}(\tilde{\mathcal{L}}(\mathbf{A}) - \tilde{\mathcal{L}}^*), \quad (74)$$

where $\tilde{\mathcal{L}}^*$ is the minimum value of $\tilde{\mathcal{L}}$. Note that $\tilde{\mathcal{L}}^*$ is exactly same to the minimum value of $\mathcal{L}$, denoted by $\mathcal{L}^*$. And so, we obtain the inequality from the Eqs. (73) and (74):

$$\frac{1}{2}m^4\|\nabla_{\mathbf{R}}\mathcal{L}\|_F^2 \geq \frac{1}{2}\|\nabla_{\mathbf{A}}\tilde{\mathcal{L}}\|_F^2 \geq \tilde{\mu}(\tilde{\mathcal{L}}(\mathbf{A}) - \tilde{\mathcal{L}}^*) = \tilde{\mu}(\mathcal{L}(\mathbf{R}) - \mathcal{L}^*). \tag{75}$$

Therefore, $\mathcal{L}$ satisfies the Polyak-Lojasiewicz condition with respect to $\mathbf{R}$:

$$\frac{1}{2}\|\nabla_{\mathbf{R}}\mathcal{L}\|_F^2 \geq \frac{\tilde{\mu}}{m^4}(\mathcal{L}(\mathbf{R}) - \mathcal{L}^*). \tag{76}$$

$\square$

Combining Lemmas E.3 and E.4, we conclude our loss function $\mathcal{L}$ converges linearly with a convergence rate $r \in (0, 1)$:

$$r = 1 - \frac{2\tilde{\mu}}{m^4(\alpha^2 + 2\alpha^3\beta)}. \tag{77}$$

Notably, the practical design choices for our model's implementation—specifically, the structured correlation matrix and the label smoothing technique—play a pivotal role in these proofs. This demonstrates a strong coherence between the practical design of our methodology and its theoretical underpinnings, suggesting that they are mutually reinforcing.

## F    MORE DETAILS ON EXPERIMENTS

**Configurations.** To ensure the reproducibility of scalability and performance, we specify our key software and hardware configurations. We implement our model using PyTorch 2.7 with CUDA 12.4. The majority of experiments are conducted on an NVIDIA A6000 GPU (48GB VRAM), with the exception of the ablation studies, which are performed on an NVIDIA A5000 GPU (24GB VRAM).

**Dataset and Preprocessing.** To make sure our experiments are fair and can be replicated, we include our entire preprocessing pipeline in our code, which works directly with the raw data files from their original sources. Our preprocessing process has two main steps. First, we treat the data as an undirected graph. If we find conflicting signs for reciprocal edges between two nodes in the raw data, we clear up this confusion by giving the edge a negative sign. This is based on the fact that negative edges are much less common across all datasets. Second, we pull out the largest connected component from each graph. The original graphs usually have one main component along with several smaller, disconnected subgraphs. We consider these smaller components as noise and leave them out—this is a common approach in graph-based research that helps us focus on a structurally coherent subgraph. You can find the statistics of the preprocessed datasets we used for our experiments in Tab. I.

Table I: Statistics of the preprocessed datasets.

| Dataset | Nodes | Positive Edges | Negative Edges | Positive Ratio |
|---|---|---|---|---|
| BitcoinAlpha | 3,775 | 12,721 | 1,399 | 90.09 (%) |
| BitcoinOTC | 5,875 | 18,230 | 3,259 | 84.83 (%) |
| WikiElec | 7,066 | 78,441 | 22,253 | 77.90 (%) |
| WikiRfa | 11,259 | 132,187 | 39,415 | 77.03 (%) |
| SlashDot | 82,140 | 380,933 | 119,548 | 76.11 (%) |
| Epinions | 119,130 | 583,957 | 120,462 | 82.90 (%) |

**Baselines.** Graph Convolutional Networks (Kipf & Welling, 2017; Gilmer et al., 2017) laid the foundation for message passing on graphs by aggregating neighborhood information under the homophily assumption. However, this assumption is violated in signed graphs due to the presence of negative edges. To address this, SGCN (Derr et al., 2018) generalized GCN based on balance theory, and SDGNN (Huang et al., 2021) extended this idea to directed signed graphs by incorporating social theories into aggregation and loss functions.

With the growing interest in attention mechanisms, SNEA (Li et al., 2020) adapts GAT (Velickovic et al., 2018) to signed graphs, enabling the model to learn separate attention weights for positive and negative neighbors. To the best of our knowledge, SLGNN (Li et al., 2023) represents the current state-of-the-art in signed graph neural networks, combining graph spectral filtering and SGNNs. Its self-gating mechanism adaptively captures information from both balanced and imbalanced structures.

Other approach is introduced by TrustSGCN (Kim et al., 2023), which incorporated trustworthiness measures to weight message passing more reliably than using raw edge signs alone. Beyond performance-oriented approaches, recent works such as SE-SGformer (Li et al., 2025) and SGAAE (Nakis et al., 2025) explore explainability by leveraging Transformer architectures and auto-encoding techniques to better interpret signed network behaviors.

Nevertheless, a common limitation across these methods is their separate handling of positive and negative edges during aggregation, which overlooks the statistical dependencies that may exist between them. Motivated by this, we propose to directly model these inter-edge relationships by learning a correlation matrix. This approach not only provides a more holistic view of the graph structure but also, as our results show, leads to a dramatic acceleration in model convergence.

**Details of Experimental Setting.** One of the major hurdles in the field of link sign prediction is the absence of a standardized evaluation protocol. Previous studies often vary in crucial elements that affect performance, such as how data is split and how features are managed, which makes it tough to make fair comparisons. To tackle this problem, we unify experimental options to reassess baseline models using their official implementations. This protocol is guided by the following principles:

- **Node features:** To simulate real-world node attributes, which are often absent in publicly available signed graph datasets, we initialize node features from a standard normal distribution without scaling. Consistent with this simulation of pre-existing attributes, these features remain fixed (*i.e.*, non-learnable) throughout the training process.

- **Data split:** We establish a robust evaluation protocol to ensure the reliability and generalizability of our results. First, we partition the data into training, validation, and test sets using an 8:1:1 ratio. This deviates from the 8:2 split used in some prior work by incorporating a dedicated validation set, which is crucial for systematic hyperparameter tuning and early stopping to mitigate overfitting. Furthermore, to account for performance variance arising from data splits, we conduct each experiment on 10 distinct data splits. The final reported metrics are the average of these runs. This multi-run evaluation provides a more stable and accurate assessment of model generalization, a step that is particularly vital for smaller datasets like BitcoinAlpha and BitcoinOTC where results can be sensitive to the specific data split.

- **Linear probing:** Current research in link sign prediction is broadly divided into two paradigms: end-to-end frameworks with integrated classifiers (Derr et al., 2018; Li et al., 2023; 2025), and two-stage approaches that first learn node embeddings for evaluation with a separate classifier (Li et al., 2020; Huang et al., 2021). This methodological divergence presents a challenge for direct and fair performance comparisons. To address this, we adopt a standardized evaluation protocol. For end-to-end models, we utilize their native classifiers as designed. For all two-stage, embedding-based methods, we standardize the downstream task by employing a single, fixed logistic regression classifier from the scikit-learn library to evaluate the quality of the learned representations.

### F.1 Additional Experimental Results

**Embedding Size Study.** We report the effect of different embedding sizes $d$ on model performance and scalability in Fig. I. As expected, a larger embedding size tends to demand more computational resources (*e.g.*, training time per epoch, inference time, and GPU memory). However, a larger embedding size does not necessarily increase the total training time, as it often leads to faster convergence in fewer epochs. Although the model remains robust to variations in $d$ overall, instances of underfitting or overfitting are observed in specific datasets where $d$ is extremely small or large, respectively.

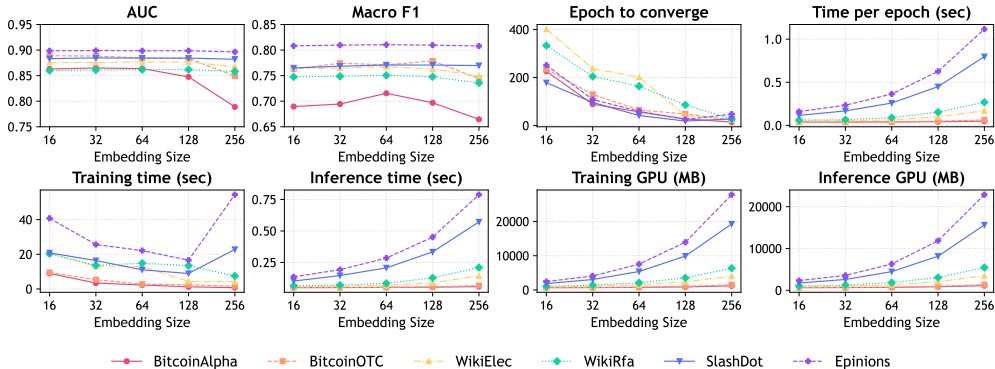

Figure I: Embedding size study of CopulaLSP.

**Performance and Convergence Study on $\epsilon$.** We investigate the sensitivity of the performance and convergence with respect to variations in $\epsilon$. As indicated in these tables and visualized in Fig. II, the model maintains consistent performance across a broad range of parameter values, confirming its robustness. As shown in Fig. III, no discernible trend is observed in convergence speed concerning changes in $\epsilon$. The optimal hyperparameter configurations for each dataset are detailed in Tab. II.

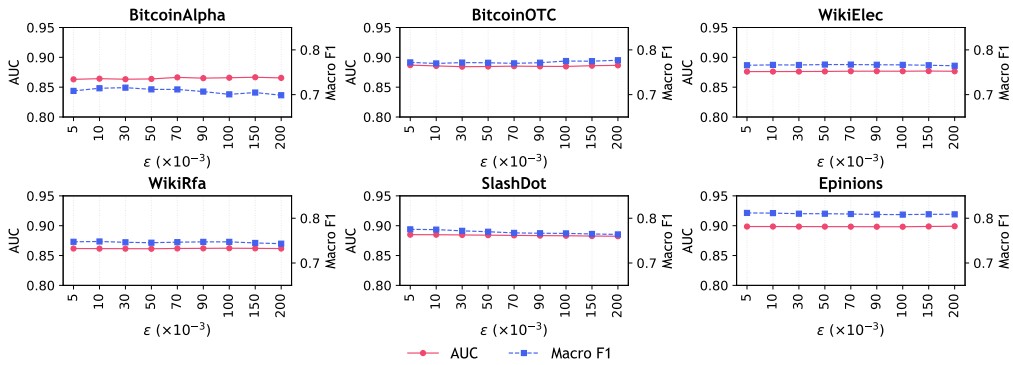

Figure II: Performance of CopulaLSP varying hyperparameter $\epsilon$.

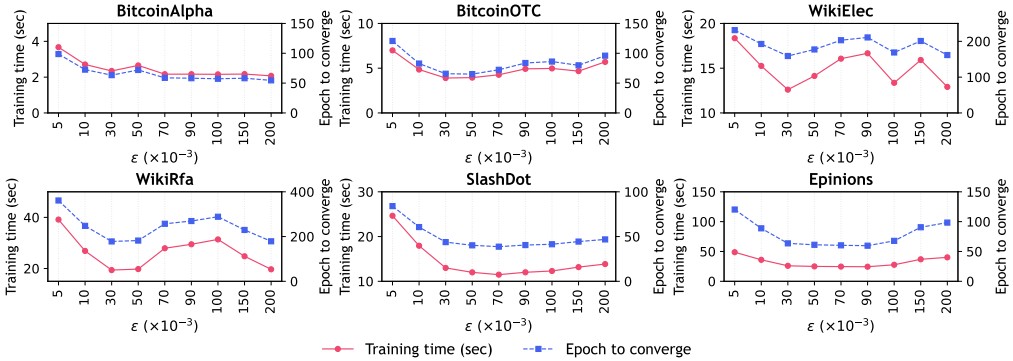

Figure III: Convergence of CopulaLSP varying hyperparameter $\epsilon$.

Table II: Best hyperparameters $\epsilon$ and $\eta$ for CopulaLSP.

| Dataset | Best $\epsilon$ | Best $\eta$ |
|---|---|---|
| BitcoinAlpha | 0.04 | 0.0008 |
| BitcoinOTC | 0.05 | 0.0001 |
| WikiElec | 0.04 | 0.005 |
| WikiRfa | 0.04 | 0.01 |
| SlashDot | 0.04 | 0.01 |
| Epinion | 0.04 | 0.001 |

**Scalability Analysis on Other Datasets.** To demonstrate the robustness of our method beyond the Wiki dataset analyzed in the main text, we present scalability comparisons for other datasets.

Table III: Overall time and memory efficiency comparison on BitcoinAlpha and BitcoinOTC.

| | BitcoinAlpha | | | BitcoinOTC | | |
|---|---|---|---|---|---|---|
| | Training (sec) | Inference (sec) | GPU (GB) | Training (sec) | Inference (sec) | GPU (GB) |
| GCN | 5.3 | 2.43 | 0.61 | 6.86 | 3.12 | **0.68** |
| SGCN | 136.3 | 0.07 | 0.62 | 341.46 | 0.07 | 0.72 |
| SNEA | 131.2 | 1.12 | **0.60** | 174.9 | 1.54 | **0.68** |
| SDGNN | 9.4 | 3.99 | 0.84 | 12.81 | 4.75 | 1.12 |
| TrustSGCN | 161.0 | 2.22 | 1.22 | 322.84 | 4.52 | 1.31 |
| SLGNN | 23.8 | **0.02** | 0.72 | 27.34 | **0.03** | 0.81 |
| SGAAE | 14.5 | 0.09 | 0.77 | 16.76 | 0.13 | 1.13 |
| SE-SGformer | 236.8 | 0.48 | 2.13 | 523.19 | 0.40 | 4.51 |
| CopulaLSP (ours) | **3.0** | 0.07 | 0.74 | **3.6** | 0.07 | 0.82 |

Table IV: Overall time and memory efficiency comparison on SlashDot and Epinions. OOM indicates out-of-memory.

| | SlashDot | | | Epinions | | |
|---|---|---|---|---|---|---|
| | Training (sec) | Inference (sec) | GPU (GB) | Training (sec) | Inference (sec) | GPU (GB) |
| GCN | 73.0 | 5.90 | 5.24 | 237.80 | 16.60 | 7.17 |
| SGCN | 5557.4 | **0.22** | 5.72 | 15008.90 | **0.28** | 8.48 |
| SNEA | 4676.5 | 5.35 | **4.96** | 6574.4 | 6.98 | **7.03** |
| SDGNN | OOM | OOM | OOM | OOM | OOM | OOM |
| TrustSGCN | OOM | OOM | OOM | OOM | OOM | OOM |
| SLGNN | 281.8 | 0.23 | 33.38 | OOM | OOM | OOM |
| SGAAE | OOM | OOM | OOM | OOM | OOM | OOM |
| SE-SGformer | OOM | OOM | OOM | OOM | OOM | OOM |
| CopulaLSP (ours) | **12.4** | 0.23 | 5.08 | **24.7** | 0.31 | 7.20 |

## F.2 MODEL ANALYSIS ON SYNTHETIC DATASET

**Model Performance on Synthetic Dataset.** To demonstrate the superiority of CopulaLSP over node-centric models, we utilize a synthetic dataset comprising 40 nodes divided into two distinct communities, as illustrated in Fig. IV. Nodes 0–19 constitute Group 1, while nodes 20–39 constitute Group 2. The graph is characterized by strong intra-group cohesion and inter-group hostility; specifically, it contains 183 and 178 intra-group positive edges for Group 1 and Group 2, respectively, alongside 76 inter-group negative edges. The edge indices are sorted such that intra-group positive edges appear first, followed by inter-group negative edges.

We train both our method and SNEA on this dataset for 10 epochs using an 8:2 train-test split. The experimental results reveal a distinct contrast. SNEA predicts all edges as positive, failing to identify inter-group hostility, whereas our method achieves perfect scores in both AUC and F1. This performance discrepancy stems from the node-centric mechanism of SNEA, which generates embeddings by aggregating topological information from neighbors. Since both groups possess nearly identical intra-group structures, nodes in opposing groups generate virtually indistinguishable embeddings despite being connected by negative edges. Consequently, SNEA fails to discern negative edges within such structurally symmetric configurations. In contrast, our method effectively resolves this ambiguity by modeling the signs of individual edges via marginal probability distributions and explicitly capturing the correlations between edges.

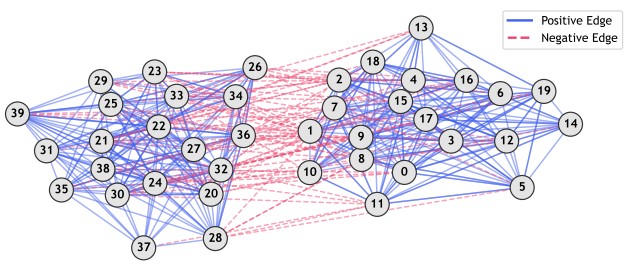

Figure IV: Topology of the synthetic signed graph comprising two symmetric communities.

**Model Component Analysis on Synthetic Dataset.** To elucidate how CopulaLSP successfully distinguishes edge signs where baseline models fail, we visualize the learned components—edge embeddings, marginal probability distributions, and the correlation matrix—in Fig. V.

First, the PCA visualization of edge embeddings demonstrates a clear separation between positive and negative edges. Notably, the positive embeddings form two distinct yet symmetrical clusters, accurately mirroring the underlying topology where two groups possess identical structures but distinct community memberships.

Second, the analysis of the location parameter $a \in (0, \infty)$ validates the efficacy of the relaxed Bernoulli distribution. We observe that $a$ is generally greater than 1 for positive edges and less than 1 for negative edges. Although a small subset of edges exhibits parameters misaligned with their actual signs, our model effectively mitigates these local discrepancies at inference. This robustness is achieved by incorporating the temperature parameter $t \in (0, 1)$ and, more importantly, by leveraging explicit inter-edge correlations to compensate for individual uncertainties.

Finally, the estimated correlation matrix provides insight into how the model captures structural dependencies. Given the edge ordering (intra-Group 1, intra-Group 2, and inter-group edges), the matrix exhibits strong positive correlations within the diagonal blocks, confirming that edges within the same structural category share high mutual information. Interestingly, we observe weak positive correlations between the blocks of Group 1 and Group 2; this reflects that, despite belonging to conflicting communities, the two groups share topologically identical structures. Crucially, the correlations between intra-group (positive) and inter-group (negative) edges are predominantly negative. This indicates that CopulaLSP successfully learns to distinguish edges that share common nodes but possess opposing semantic meanings, thereby resolving the ambiguity that confounds node-centric models.

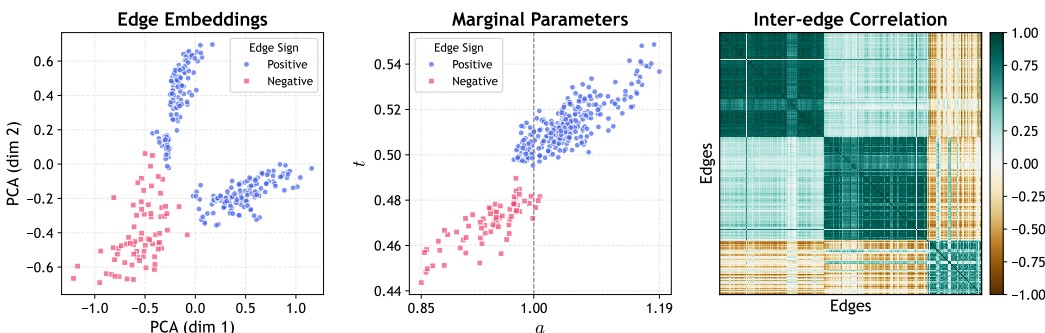

Figure V: Component analysis of CopulaLSP on synthetic dataset.

