# OpenReview forum: "A Scalable Inter-edge Correlation Modeling in CopulaGNN for Link Sign Prediction"
_ICLR.cc/2026/Conference — ICLR 2026 Poster_

### Official Review · Reviewer_87t8 · 2025-10-22

**Soundness:** 3
**Presentation:** 3
**Contribution:** 3
**Rating:** 6
**Confidence:** 3

**Summary:**

This study proposes CopulaLSP, which employs a Gaussian copula to couple the marginal relaxed Bernoulli distributions with the correlation structure derived from the Gramian matrix of edge embeddings. It introduces the Woodbury reformulation to maximize the spatiotemporal efficiency of sampling-based inference and theoretically validates the rapid convergence of the method.

**Strengths:**

The study features rigorous theoretical analysis, supplemented by visual illustrations to elucidate parameter functions. The integration of theoretical and visual elements ensures clarity in demonstrating methodological robustness and practical applicability.

**Weaknesses:**

The study exhibits an excessive emphasis on theoretical analysis, while the experimental section appears somewhat rudimentary, lacking in-depth investigation into the configuration of hyperparameters.

**Questions:**

1.When using the Woodbury reformulation for matrix inversion, how stable is the inversion process when the embedding dimension is large?
2.To what extent does the setting of the label softening parameter ηimpact model performance, particularly convergence speed? While the appendix mentions a hyperparameter study, the main text lacks a sensitivity analysis of η.

---

> ### Author Response · Authors · 2025-11-23
> **Official Comment for 87t8 (1/2)**
>
> __(W1) The study exhibits an excessive emphasis on theoretical analysis, while the experimental section appears somewhat rudimentary, lacking in-depth investigation into the configuration of hyperparameters.__
>
> We appreciate the constructive feedback. Following the feedback, we have streamlined the theoretical sections to allow for a significantly reinforced empirical validation. First, to better highlight the comparative advantage of our method, the scalability comparison (Table 3) against SNEA and other state-of-the-art models has been relocated to the main text (Section 5.2) from Appendix. Second, we conducted a more rigorous hyperparameter study. We analyzed the convergence behavior and performance sensitivity with respect to $\epsilon$ and $\eta$ across an expanded suite of six datasets, including two newly added ones (Appendix F.1). Furthermore, we broadened the hyperparameter search space to verify the robustness of our method. We believe these extensive updates have significantly improved the quality of the manuscript and thoroughly addressed your concerns.
>
> ---
>
> __(Q1) When using the Woodbury reformulation for matrix inversion, how stable is the inversion process when the embedding dimension is large?__
>
> _Numerical Stability of Inversion on_ $\mathbf{S}_0$. We monitored the maximum condition number of the matrix $\mathbf{S}_0$ throughout the training process across various embedding dimensions ($d = 16, 32, 64, 128, 256$).
>
> To ensure precision, we utilized `float64` for the Woodbury reformulation computations. The table below reports the maximum condition number of $\mathbf{S}_{0}$, confirming that the condition number remained below $250,000$.
>
> Notably, no discernible pattern or correlation was observed between the embedding dimension $d$ and the magnitude of the condition number. Given that double precision provides approximately $16$ significant decimal digits, a condition number of this magnitude implies a theoretical loss of roughly $5$ to $6$ digits. This preserves about $10$ significant digits, ensuring that the inversion operation remains numerically stable.
>
> | Dataset | $d=16$ | $d=32$ | $d=64$ | $d=128$ | $d=256$ |
> | :--- | :---: | :---: | :---: | :---: | :---: |
> | **BitcoinAlpha** | 79122 | 169065 | 176791 | 128012 | 60600 |
> | **BitcoinOTC** | 63870 | 60989 | 149418 | 199528 | 63518 |
> | **WikiElec** | 13101 | 34979 | 133721 | 41050 | 45066 |
> | **WikiRfa** | 111131 | 27449 | 34169 | 75459 | 46648 |
> | **SlashDot** | 154185 | 854 | 57786 | 81111 | 134936 |
> | **Epinions** | 235536 | 46330 | 95985 | 75981 | 165316 |
>
>
>
> _Instability of Cholesky Decomposition._ For the eigenvalues $\lambda_{i}$ of $\mathbf{Q}^\top \mathbf{Q}$, the $\det \mathbf{S}_{0}$ (simply write $\mathbf{S}$) is expressed as:
>
> $$\det {\mathbf{S}}= \prod_{i=1}^{d} \left( 1 + \frac{\lambda_{i}}{\epsilon} \right).$$
>
> Since the terms are positive, $\mathbf{S}^{-1}$ theoretically always exists. However, as the dimension $d$ increases, $\det \mathbf{S}$ grows significantly, which implies that the eigenvalues of its inverse become extremely small. This scale disparity causes computational instability when performing Cholesky decomposition on $\mathbf{S}^{-1}$. To resolve this, we applied a jitter term $(1e-4) \cdot \mathbf{I}$ and performed the decomposition on $\mathbf{S}^{-1} + (1e-4) \cdot \mathbf{I}$.

---

> ### Author Response · Authors · 2025-11-23
> **Official Comment for 87t8 (2/2)**
>
> __(Q2) To what extent does the setting of the label softening parameter η impact model performance, particularly convergence speed? While the appendix mentions a hyperparameter study, the main text lacks a sensitivity analysis of η.__
>
> Increasing $\eta$ leads to a decrease in the norm of $\mathbf{z}_{obs}$.
>
> As $\eta$ increases, the smoothed labels $\bar{y}$$_{i}$ shift away from the boundaries (0 and 1) and move closer to $\frac{1}{2}$.
>
> Since the marginal CDF $F_{i}$ is a monotonically increasing function, the transformed values $u_{i} = F_{i}(\bar{y}_{i})$ also shift away from the extremes.
>
> Consequently, the inverse Gaussian CDF $\Phi^{-1}(u_{i})$ yields smaller magnitudes as the input $u_{i}$ approaches $\frac{1}{2}$, leading to a reduction in the norm of $\mathbf{z}_{obs}$.
>
> Importantly, as detailed in Appendix E, this reduction in the norm lowers the upper bound constant $\beta$, where $||\mathbf{z_\mathrm{obs}z_\mathrm{obs}}^\top || \le \beta$.
>
> A smaller $\beta$ results in a smaller convergence rate $r$ in our theoretical bound, thereby guaranteeing faster linear convergence. This theoretical analysis is consistent with the empirical results (Figure 3) that we updated in the 'Convergence Study' of Section 5.4 (Line [484-503]).

---

### Official Review · Reviewer_y8s8 · 2025-10-25

**Soundness:** 3
**Presentation:** 3
**Contribution:** 2
**Rating:** 6
**Confidence:** 3

**Summary:**

This paper introduces CopulaLSP, a scalable framework for link sign prediction in signed graphs. Based on the assumption of statistical dependencies between edges, this paper overcomes the inability of traditional GNNs to handle negative edges, extending the architecture from unsigned graphs to signed graphs. It uses a Gramian of lower-dimensional edge embeddings to model the correlation between edges, which significantly reduces the number of learnable parameters and memory consumption. Furthermore, it reformulates the conditional probability distribution using the Woodbury matrix identity, transforming the matrix inversion required during inference into the inversion of a smaller matrix, significantly reducing computational cost. The proposed method is proven to have linear convergence.

**Strengths:**

1. The paper proposes an innovative hypothesis that there exists a statistical dependence between edges connected by common nodes, thus extending GNNs from unsigned to signed graphs.

2. By introducing a Gramian-based correlation matrix for edge dependencies and a Woodbury matrix rewrite for computational efficiency, the paper significantly reduces memory usage and computational cost, greatly accelerates model convergence, and achieves good scalability while maintaining performance comparable to baseline models.

3. The paper provides rigorous mathematical support for its core claims through numerous mathematical derivations.

4. Ablation experiments on the two core components demonstrate the effectiveness of the proposed method.

5. The paper has a clear and well-organized structure, with distinct separation of different components.

**Weaknesses:**

1. The paper contains a large number of mathematical formulas, which makes it somewhat difficult to read.

2. The innovation of the framework mainly relies on the creative combination of existing tools (Gaussian Copula, Gramian construction, and the Woodbury identity), lacking some novelty.

3. Model training depends on hyperparameters η and ε, making it difficult to directly obtain optimal model performance.

4. Section 4 spends some length analyzing the model's convergence, which is not directly related to the main logic of the paper.

5. The method only uses SNEA as the backbone encoder, which raises questions about its generalizability.

6. Time and memory efficiency comparison is a key indicator for assessing the performance of the method, but the paper only provides comparisons with SNEA, making the experimental results less persuasive.

**Questions:**

1. Is the method insensitive to the backbone encoder?

---

> ### Author Response · Authors · 2025-11-23
> **Official Comment for y8s8 (1/2)**
>
> __(W1) The paper contains a large number of mathematical formulas, which makes it somewhat difficult to read.__
>
> Please refer to the answer for W4.
>
> ---
>
> __(W2) The innovation of the framework mainly relies on the creative combination of existing tools (Gaussian Copula, Gramian construction, and the Woodbury identity), lacking some novelty.__
>
> We respectfully disagree with the view that our framework lacks novelty. While we utilize established mathematical tools, the core innovation lies in how these components are strategically integrated to solve a critical bottleneck that has long plagued Signed Graph Neural Networks: the intractability of modeling inter-edge correlations.
>
> Specifically, our contribution is not merely a summation of parts but a tailored architectural design:
>
> 1. Gramian Construction: We adapted this to satisfy the positive definiteness constraint while reducing memory complexity from $O(n^2)$ to $O(nd)$.
>
> 2. Woodbury Reformulation: This was not applied arbitrarily but was necessitated by our Gramian design to shift the computational bottleneck from graph size ($n$) to embedding size ($d$), enabling the processing of a large-scale graph.
>
> 3. Theoretical Coherence: As proven in Appendix E, these choices (along with the label smoothing) are mathematically interlinked to guarantee linear convergence.
>
> We believe that orchestrating these elements to transform a theoretically desirable but computationally prohibitive concept into a scalable, working model constitutes a significant innovation.
>
> ---
>
> __(W3) Model training depends on hyperparameters η and ε, making it difficult to directly obtain optimal model performance.__
>
> We respectfully point out that, in practice, obtaining optimal performance is quite manageable for two reasons. As shown in Figure 4 of Section 5, and Figure III of Appendix F.1, the model's performance (AUC, F1) is remarkably robust to changes in $\epsilon$ and $\eta$, reducing the need for precise tuning. Moreover, the extremely fast convergence of our model drastically alleviates the burden of hyperparameter search, allowing optimal settings to be identified with minimal computational overhead compared to existing baselines.
>
> ---
>
> __(W4) Section 4 spends some length analyzing the model's convergence, which is not directly related to the main logic of the paper.__
>
> We appreciate your constructive feedback regarding the organization of the paper. While the convergence analysis in Section 4 serves as the theoretical foundation for our model's fast training speed—a core component of our main logic—we agree that the detailed proofs consumed excessive space. Therefore, we moved the detailed theoretical derivations to the Appendix E, and moved the scalability comparison (Table 3) against SNEA and other state-of-the-art models from Appendix to Section 5.2. We believe this structural change effectively rebalances the paper, reinforcing the practical value of our work while maintaining theoretical soundness.
>
> ---
>
> __(W5) The method only uses SNEA as the backbone encoder, which raises questions about its generalizability.__
>
> Please refer to the answer for Q1.
>
> ---
>
> __(W6) Time and memory efficiency comparison is a key indicator for assessing the performance of the method, but the paper only provides comparisons with SNEA, making the experimental results less persuasive.__
>
> Agreeing with the reviewer’s comment, we moved the scalability comparisons against a broader range of baselines to Table 3 in Section 5.2.

---

> ### Author Response · Authors · 2025-11-23
> **Official Comment y8s8 (2/2)**
>
> __(Q1) Is the method insensitive to the backbone encoder?__
>
> _MLP backbone test._ We conducted a comparative analysis between a pure MLP model, which predicts sign probabilities using node features as input, and our proposed CopulaLSP framework utilizing an MLP backbone (Our-MLP). For the pure MLP, we evaluated architectures ranging from 3 to 6 layers to sufficiently address potential limitations in representation capacity. In contrast, Our-MLP employed a fixed 3-layer MLP.
>
> Empirical results indicate that the pure MLP consistently underperforms across all datasets. We attribute this performance gap to the inherent limitation of the pure MLP in capturing the graph's topological structure. Conversely, Our-MLP not only achieves superior performance metrics but also demonstrates enhanced efficiency, reaching optimal performance with fewer training epochs and reduced training time.
>
> | Dataset | Model | AUC| F1 | Epoch to converge| Training time (sec) | Inference time (sec) |
> | --- | --- | ---: | ---: | ---: | ---: | ---: |
> | BitcoinAlpha | MLP-3 | 0.771 | 0.587 | 187 | 2.18 | 0.03 |
> | | MLP-4 | 0.769 | 0.622 | 147 | 1.96 | 0.03 |
> | | MLP-5 | 0.767 | 0.629 | 147 | 3.44 | 0.05 |
> | | MLP-6 | 0.757 | 0.631 | 136 | 3.61 | 0.05 |
> | | Our-MLP | 0.821 | 0.660 | 170 | 2.55 | 0.04 |
> | BitcoinOTC | MLP-3 | 0.784 | 0.659 | 223 | 2.56 | 0.03 |
> | | MLP-4 | 0.790 | 0.681 | 226 | 2.96 | 0.03 |
> | | MLP-5 | 0.785 | 0.679 | 231 | 5.42 | 0.05 |
> | | MLP-6 | 0.780 | 0.679 | 222 | 5.85 | 0.05 |
> | | Our-MLP | 0.850 | 0.720 | 261 | 3.90 | 0.04 |
> | WikiElec | MLP-3 | 0.532 | 0.493 | 2 | 0.03 | 0.04 |
> | | MLP-4 | 0.642 | 0.566 | 790 | 11.29 | 0.04 |
> | | MLP-5 | 0.751 | 0.637 | 1659 | 39.21 | 0.06 |
> | | MLP-6 | 0.741 | 0.628 | 1439 | 39.14 | 0.06 |
> | | Our-MLP | 0.836 | 0.690 | 716 | 13.60 | 0.04 |
> | WikiRfa | MLP-3 | 0.520 | 0.496 | 1 | 0.02 | 0.04 |
> | | MLP-4 | 0.582 | 0.529 | 575 | 8.49 | 0.04 |
> | | MLP-5 | 0.626 | 0.549 | 979 | 25.53 | 0.07 |
> | | MLP-6 | 0.715 | 0.594 | 1212 | 36.66 | 0.07 |
> | | Our-MLP | 0.806 | 0.651 | 860 | 18.36 | 0.05 |
> | SlashDot | MLP-3 | 0.523 | 0.505 | 1 | 0.04 | 0.06 |
> | | MLP-4 | 0.567 | 0.507 | 400 | 11.15 | 0.06 |
> | | MLP-5 | 0.628 | 0.556 | 993 | 38.47 | 0.12 |
> | | MLP-6 | 0.731 | 0.615 | 1569 | 73.08 | 0.12 |
> | | Our-MLP | 0.794 | 0.665 | 989 | 37.85 | 0.08 |
> | Epinions | MLP-3 | 0.528 | 0.508 | 2 | 0.07 | 0.07 |
> | | MLP-4 | 0.726 | 0.622 | 1597 | 59.24 | 0.07 |
> | | MLP-5 | 0.767 | 0.649 | 1757 | 90.25 | 0.14 |
> | | MLP-6 | 0.761 | 0.647 | 1637 | 101.77 | 0.15 |
> | | Our-MLP | 0.808 | 0.670 | 913 | 46.30 | 0.09 |
>
> _SLGNN backbone test._ To demonstrate that our proposed framework is not limited to a specific architecture, we evaluated a variant of our model, denoted as Ours-SLGNN, where the SNEA backbone is replaced with SLGNN. Experimental results across six datasets show that Ours-SLGNN maintains predictive performance comparable to the original SLGNN while significantly reducing both the number of epochs required for convergence and the total training time. This finding, combined with the results from the MLP backbone experiments, strongly supports the conclusion that our correlation modeling approach generalizes effectively across different underlying architectures.
>
> However, we observed that the complex architecture of SLGNN incurs a substantial memory footprint; specifically, it consumes approximately 30GB of GPU memory on the SlashDot dataset and fails to operate on the larger Epinions dataset due to out-of-memory errors. Since our primary design objective was to ensure scalability and to facilitate deployment in real-world applications, we intentionally selected the more lightweight SNEA as our primary backbone.
>
> | Dataset | Model | AUC | F1 | Epoch to converge | Training time (sec) | Inference time (sec) |
> | --- | --- | ---: | ---: | ---: | ---: | ---: |
> | **BitcoinAlpha** | SLGNN | 0.864 | 0.716 | 1936 | 34.3 | 0.03 |
> | | Our-SLGNN | 0.871 | 0.706 | 143 | 8.8 | 0.06 |
> | **BitcoinOTC** | SLGNN | 0.903 | 0.818 | 1953 | 27.5 | 0.03 |
> | | Our-SLGNN | 0.912 | 0.806 | 107 | 6.2 | 0.08 |
> | **WikiElec** | SLGNN | 0.887 | 0.770 | 1975 | 64.6 | 0.06 |
> | | Our-SLGNN | 0.887 | 0.763 | 323 | 27.3 | 0.08 |
> | **WikiRfa** | SLGNN | 0.874 | 0.748 | 1972 | 78.9 | 0.08 |
> | | Our-SLGNN | 0.870 | 0.738 | 151 | 13.1 | 0.11 |
> | **SlashDot** | SLGNN | 0.890 | 0.771 | 1953 | 281.8 | 0.23 |
> | | Our-SLGNN | 0.892 | 0.772 | 67 | 68.9 | 0.25 |

---

### Official Review · Reviewer_6GuR · 2025-10-27

**Soundness:** 3
**Presentation:** 3
**Contribution:** 2
**Rating:** 6
**Confidence:** 3

**Summary:**

The paper proposes CopulaGNN for Link Sign Prediction (CopulaLSP). It builds on the prior work CopulaGNN for the task of node regression, the core idea of which is the model dependencies between nodes by learning a Gaussian copula. Here the model must instead learn correlations between edges, which is intractable with a naïve approach, so the authors propose two ideas to deal with this. First, the correlation matrix is parametrized as the Gramian of edge embeddings (which are learned with a prior GNN model for signed graphs, SNEA). Second, a Woodbury matrix identity is used to transform the inference-time inversion of a matrix of the size of the number of observed edges, to the inversion of the matrix of the size of the embedding dimension. Furthermore, the authors prove that their approach converges linearly. Finally, the method is evaluated against other recent methods on 4 real-world datasets, showing competitive prediction performance and faster convergence.

**Strengths:**

- The paper has a clear and significant narrative: It considers the natural approach of edge-edge correlation modeling, which has some general relevance in graph modeling, identifies the scalability problem, and provides a solution.
- The writing is clear and grammatical. The diagram of Figure 1 is helpful for understanding the core concepts.
- The paper includes conceptual, theoretical, and experimental components.
- Several ablations are provided to strengthen the paper's claims.

**Weaknesses:**

- The method is competitive with others in terms of prediction performance, but not clearly superior and arguably inferior to one other method on the chosen datasets.
- The diversity of the datasets is limited, with two Bitcoin datasets and two Wikipedia datasets.
- There is little discussion of how the prior methods work, including SNEA, which is used as the encoder backbone in this paper.
- There is little discussion of the modeling side of the proposed approach, e.g., what do the learned embeddings, linear projection, and Gramian look like, on real or toy data?
- The convergence result is welcome, but it seems to simply be an application of a prior result to the proposed loss function. I would favor more space in the main body of the paper on the prior two points.

**Questions:**

- A "relaxed Bernoulli distribution" is defined in Eq. 6 as a continuous analog of the Bernoulli distribution. Has this distribution been studied before? Why are other, more commonly-used distributions on $(0,1)$, like the beta distribution, not suitable?
- Have you evaluated using the graph Laplacian of the line graph for edge correlations, as an analog to use of the graph Laplacian itself in CopulaGNN?
- The question about the modeling side of the approach above.
- How does CopulaLSP work with other backbones than SNEA? What is preventing applying the CopulaLSP on top of the strongest prior method, SLGNN? Relatedly, what would be the performance if dropping the GNN encoder backbones and learning a simpler encoder (e.g., just a linear transformation or MLP of the node features, or directly learning node embeddings if there are no node features)?

---

> ### Author Response · Authors · 2025-11-23
> **Official Comment for 6GuR (1/3)**
>
> __(W1) The method is competitive with others in terms of prediction performance, but not clearly superior and arguably inferior to one other method on the chosen datasets.__
>
> We acknowledge the reviewer’s comment that our method does not significantly outperform competing methods. We would like to clarify that the primary objective of our work is to resolve the prohibitive computational bottlenecks of direct correlation modeling, significantly improving the efficiency with minimal impact to the accuracy. We believe that achieving drastic improvements in scalability without compromising prediction accuracy constitutes a significant and practical contribution to the field.
>
> ---
>
> __(W2) The diversity of the datasets is limited, with two Bitcoin datasets and two Wikipedia datasets.__
>
> To address this, we additionally evaluate on two large-scale datasets, SlashDot and Epinions. As reported in Tab. 2 and 3 in Sec. 5.2, our proposed method demonstrates superior scalability and link sign prediction accuracy, while many other previous methods suffer from out-of-memory on these large datasets.
>
> ---
>
> __(W3) There is little discussion of how the prior methods work, including SNEA, which is used as the encoder backbone in this paper.__
>
> As per your suggestion, we have added a brief description of the SNEA architecture in the Experimental Setup (Section 5.1). For descriptions of other prior methods, please refer to the “Baseline” paragraph in Appendix F.
>
> ---
>
> __(W4) There is little discussion of the modeling side of the proposed approach, e.g., what do the learned embeddings, linear projection, and Gramian look like, on real or toy data?__
>
> We agree that a deeper look into the model's internal mechanics provides valuable intuition. In response, we added a comprehensive analysis of the learned components using a synthetic dataset in Appendix F.2.
>
> This dataset comprises two conflicting communities characterized by strong internal cohesion and external hostility,  as illustrated in Fig. IV. In this symmetric setting, node-centric baselines (e.g., SNEA) fail to capture the inter-group hostility due to the structural equivalence of the nodes. In contrast, our edge-centric approach successfully resolves this ambiguity, achieving a perfect classification score.
>
> To demonstrate how our model achieves this, we visualized and analyzed the specific components the reviewer asked in Fig. V in Appendix F.2:
>
> 1. Learned embeddings visualized in Fig. V (left) reveals clear, symmetrical clusters that distinctly separate positive and negative edges, accurately reflecting the underlying topology.
>
> 2. We verified that the linearly projected location $a$ and temperature $t$ in Fig. V (middle) align with the ground-truth edge signs, validating our use of the relaxed Bernoulli distribution.
>
> 3. Gramian: Our Inter-edge correlation matrix in Fig. V (right) explicitly captures the negative correlations between intra-group (positive) and inter-group (negative) edges. This confirms that the model learns the structural dependencies required to distinguish conflicting communities.
>
>
> ---
>
> __(W5) The convergence result is welcome, but it seems to simply be an application of a prior result to the proposed loss function. I would favor more space in the main body of the paper on the prior two points.__
>
> We appreciate this constructive feedback regarding the balance between the theoretical analysis and empirical validation. While we acknowledge that our convergence proof builds upon existing theoretical frameworks, we would like to respectfully point out that our derivation is not merely a direct application. As detailed in Appendix E, the specific design choices of our model (Gramian-based correlation structure and label smoothing) serve as essential conditions for satisfying the Polyak-Łojasiewicz (PL) condition and ensuring linear convergence.
>
> Nevertheless, we agree with your suggestion to prioritize the empirical results in the main body. Accordingly, we have streamlined the theoretical presentation in the main text to allocate more space for the following key updates:
>
> 1. We moved the comprehensive _scalability comparison against SNEA_ and other state-of-the-art models from Appendix to the main Section 5.2.
>
> 2. We significantly expanded our _hyperparameter analysis_. We investigated the convergence behavior and performance sensitivity with respect to $\epsilon$ and $\eta$ across six datasets (including two newly added ones) over a broadened search space.
>
> We believe these revisions effectively address your concerns and substantially strengthen the contributions of our work.

---

> ### Author Response · Authors · 2025-11-23
> **Official Comment for 6GuR (2/3)**
>
> __(Q1) A "relaxed Bernoulli distribution" is defined in Eq. 6 as a continuous analog of the Bernoulli distribution. Has this distribution been studied before? Why are other, more commonly-used distributions on (0, 1), like the beta distribution, not suitable?__
>
> The _relaxed Bernoulli distribution_ itself has been studied before; its Probability Density Function (PDF) is formally defined in the appendix of the _a continuous relaxation_ paper [1], which we cited in our work. As stated in the paper, our requirement for a computationally efficient and theoretically sound application of Gaussian Copulas necessitated a distribution with _closed-form expressions_ for both its Cumulative Distribution Function (CDF) and its inverse (ICDF). The Beta distribution presents a significant challenge in this regard, as it lacks these closed-form solutions. Resorting to numerical integration to compute them is computationally inefficient. Furthermore, the lower interpretability of the Beta distribution's parameters compared to those of the relaxed Bernoulli distribution was another key factor in our decision against its use.
>
>
> ---
>
> __(Q2) Have you evaluated using the graph Laplacian of the line graph for edge correlations, as an analog to use of the graph Laplacian itself in CopulaGNN?__
>
> We acknowledge this is an excellent point. Utilizing a signed graph Laplacian to derive a positive-definite precision matrix, and subsequently designing a correlation matrix, is indeed a valid design choice. However, primarily focusing on scalability, we were concerned that the process of constructing a line graph (which would be necessary for such an approach) is not only computationally prohibitive but would also result in a number of learnable parameters proportional to the number of edges. Moreover, an approach via the signed graph Laplacian does not inherently satisfy the low-rank correlation condition, which would preclude a low-rank Gaussian and thus risk significantly increasing computational complexity. Given these scalability concerns, we chose to construct a simpler yet effective model using a Gramian matrix.
>
> ---
>
> [1] Maddison, Chris J., Andriy Mnih, and Yee Whye Teh. "The concrete distribution: A continuous relaxation of discrete random variables." ICLR (2017).

---

> ### Author Response · Authors · 2025-11-23
> **Official Comment for 6GuR (3/3)**
>
> __(Q3) How does CopulaLSP work with other backbones than SNEA? What is preventing applying the CopulaLSP on top of the strongest prior method, SLGNN? Relatedly, what would be the performance if dropping the GNN encoder backbones and learning a simpler encoder (e.g., just a linear transformation or MLP of the node features, or directly learning node embeddings if there are no node features)?__
>
> _MLP backbone test._ We conducted a comparative analysis between a pure MLP model, which predicts sign probabilities using node features as input, and our proposed CopulaLSP framework utilizing an MLP backbone (Our-MLP). For the pure MLP, we evaluated architectures ranging from 3 to 6 layers to sufficiently address potential limitations in representation capacity. In contrast, Our-MLP employed a fixed 3-layer MLP.
>
> Empirical results indicate that the pure MLP consistently underperforms across all datasets. We attribute this performance gap to the inherent limitation of the pure MLP in capturing the graph topological structure. Conversely, Our-MLP not only achieves superior performance metrics but also demonstrates enhanced efficiency, reaching optimal performance with fewer training epochs and reduced training time.
>
> Furthermore, as suggested, we experimented with an approach that directly learns node embeddings. However, this method failed to converge on any dataset, with AUC and Macro-F1 scores oscillating around 0.5, effectively indicating random guessing performance.
>
> | Dataset | Model | AUC| F1 | Epoch to converge| Training time (sec) | Inference time (sec) |
> | :--- | --- | ---: | ---: | ---: | ---: | ---: |
> | BitcoinAlpha | MLP-3 | 0.771 | 0.587 | 187 | 2.18 | 0.03 |
> | | MLP-4 | 0.769 | 0.622 | 147 | 1.96 | 0.03 |
> | | MLP-5 | 0.767 | 0.629 | 147 | 3.44 | 0.05 |
> | | MLP-6 | 0.757 | 0.631 | 136 | 3.61 | 0.05 |
> | | Our-MLP | 0.821 | 0.660 | 170 | 2.55 | 0.04 |
> | BitcoinOTC | MLP-3 | 0.784 | 0.659 | 223 | 2.56 | 0.03 |
> | | MLP-4 | 0.790 | 0.681 | 226 | 2.96 | 0.03 |
> | | MLP-5 | 0.785 | 0.679 | 231 | 5.42 | 0.05 |
> | | MLP-6 | 0.780 | 0.679 | 222 | 5.85 | 0.05 |
> | | Our-MLP | 0.850 | 0.720 | 261 | 3.90 | 0.04 |
> | WikiElec | MLP-3 | 0.532 | 0.493 | 2 | 0.03 | 0.04 |
> | | MLP-4 | 0.642 | 0.566 | 790 | 11.29 | 0.04 |
> | | MLP-5 | 0.751 | 0.637 | 1659 | 39.21 | 0.06 |
> | | MLP-6 | 0.741 | 0.628 | 1439 | 39.14 | 0.06 |
> | | Our-MLP | 0.836 | 0.690 | 716 | 13.60 | 0.04 |
> | WikiRfa | MLP-3 | 0.520 | 0.496 | 1 | 0.02 | 0.04 |
> | | MLP-4 | 0.582 | 0.529 | 575 | 8.49 | 0.04 |
> | | MLP-5 | 0.626 | 0.549 | 979 | 25.53 | 0.07 |
> | | MLP-6 | 0.715 | 0.594 | 1212 | 36.66 | 0.07 |
> | | Our-MLP | 0.806 | 0.651 | 860 | 18.36 | 0.05 |
> | SlashDot | MLP-3 | 0.523 | 0.505 | 1 | 0.04 | 0.06 |
> | | MLP-4 | 0.567 | 0.507 | 400 | 11.15 | 0.06 |
> | | MLP-5 | 0.628 | 0.556 | 993 | 38.47 | 0.12 |
> | | MLP-6 | 0.731 | 0.615 | 1569 | 73.08 | 0.12 |
> | | Our-MLP | 0.794 | 0.665 | 989 | 37.85 | 0.08 |
> | Epinions | MLP-3 | 0.528 | 0.508 | 2 | 0.07 | 0.07 |
> | | MLP-4 | 0.726 | 0.622 | 1597 | 59.24 | 0.07 |
> | | MLP-5 | 0.767 | 0.649 | 1757 | 90.25 | 0.14 |
> | | MLP-6 | 0.761 | 0.647 | 1637 | 101.77 | 0.15 |
> | | Our-MLP | 0.808 | 0.670 | 913 | 46.30 | 0.09 |
>
>
> _SLGNN backbone test._ To demonstrate that our proposed framework is not limited to a specific architecture, we evaluated a variant of our model, denoted as Ours-SLGNN, where the SNEA backbone is replaced with SLGNN. Experimental results across six datasets show that Ours-SLGNN maintains predictive performance comparable to the original SLGNN while significantly reducing both the number of epochs required for convergence and the total training time. This finding, combined with the results from the MLP backbone experiments, strongly supports the conclusion that our correlation modeling approach generalizes effectively across different underlying architectures.
>
> However, we observed that the complex architecture of SLGNN incurs a substantial memory usage; specifically, it consumes approximately 30GB of GPU memory on the SlashDot dataset and fails to operate on the larger Epinions dataset due to out-of-memory errors. Since our primary design objective was to ensure scalability and to facilitate deployment in real-world applications, we intentionally selected the more lightweight SNEA as our primary backbone.
> | Dataset | Model | AUC | F1 | Epoch to converge | Training time (sec) | Inference time (sec) |
> | :--- | :--- | ---: | ---: | ---: | ---: | ---: |
> | **BitcoinAlpha** | SLGNN | 0.864 | 0.716 | 1936 | 34.3 | 0.03 |
> | | Our-SLGNN | 0.871 | 0.706 | 143 | 8.8 | 0.06 |
> | **BitcoinOTC** | SLGNN | 0.903 | 0.818 | 1953 | 27.5 | 0.03 |
> | | Our-SLGNN | 0.912 | 0.806 | 107 | 6.2 | 0.08 |
> | **WikiElec** | SLGNN | 0.887 | 0.770 | 1975 | 64.6 | 0.06 |
> | | Our-SLGNN | 0.887 | 0.763 | 323 | 27.3 | 0.08 |
> | **WikiRfa** | SLGNN | 0.874 | 0.748 | 1972 | 78.9 | 0.08 |
> | | Our-SLGNN | 0.870 | 0.738 | 151 | 13.1 | 0.11 |
> | **SlashDot** | SLGNN | 0.890 | 0.771 | 1953 | 281.8 | 0.23 |
> | | Our-SLGNN | 0.892 | 0.772 | 67 | 68.9 | 0.25 |

---

### Official Review · Reviewer_bz2b · 2025-10-28

**Soundness:** 4
**Presentation:** 4
**Contribution:** 2
**Rating:** 6
**Confidence:** 4

**Summary:**

The authors study a problem of graph learning on signed graph, where existing GNNs struggle due to the homophily assumption. Thus, the authors borrow the idea from CopulaGNN. However, directly applying CopulaGNN is not applicable due to computational limitations. Two main ideas were introduced to tackle the limitations. First, (at the training phase) the correlation matrix is constructed as a Gramian of edge embeddings, which dramatically reduces the number of parameters. Second, the conditional probability distribution has been reformulated for reducing the inference cost. Experimental results show how CopulaLSP achieves significant reductions in computation time while maintaining the predictive performances from competitive models.

**Strengths:**

- The main problem/ task is one of the important topic in GNN. The problem formulation (in Section 2) is clear. The paper is well organized and relevant theorem has been provided to support the statements.
- The main idea: using the CopulaGNN for signed graph learning has been well justified theoretically. The idea of reducing the computational cost in two ways: training phase and inference phase, is straightforward and supported well.
- For the experimental results, the computational reduction, both in time and memory, is significant. Ablation study was conducted thoroughly, which supports the efficacy of CopulaLSP.

**Weaknesses:**

- Some of the details are missing requiring further clarifications.(see questions)
- How does the low-rank multivariate Gaussian brings, in what extent, how significant ? The authors simply state that Woodbury reformulation is used to improve computational efficiency.
- The proposed model uses SNEA as their backbone and show the improvements in computational efficiency. However, from practical point, considering that the best prediction performances are achieved among TrustSGCN, SLGNN, further computational comparison in the Appendix can be added in Section 5. In some sense, the achievement (437 times faster convergence than baseline) can be viewed as an overstatement considering that the baseline is SNEA.

**Questions:**

Q1. The notion of missing edges can be further clarified. In conventional GNN or in graph learning models, missing edges refer to unobserved relationships.

Q2. I wonder how their ‘label softening’ is different from ‘label smoothing’ which is well-established term.

Q3. Can you further elaborate why the correlation matrix should satisfy the condition in line 180-181? It is unclear why this condition should be added supposedly matrix R is the same from line 127-128 which is directly borrowed from CopulaGNN.

---

> ### Author Response · Authors · 2025-11-23
> **Official Comment for bz2b (1/2)**
>
> __(W1) How does the low-rank multivariate Gaussian brings, in what extent, how significant? The authors simply state that Woodbury reformulation is used to improve computational efficiency.__
>
> _A Low-rank Gaussian Distribution_ refers to a multivariate Gaussian distribution whose covariance matrix is expressed in the form of $\mathbf{AA}^\top + \mathbf{B}$, where $\mathbf{A}$ is a covariance factor matrix and $\mathbf{B}$ is a covariance diagonal matrix. This low-rank covariance structure allows computationally efficient implementation of our algorithm using the Woodbury matrix identity. For details, refer to the `LowRankMultivariateNormal` class in PyTorch.
>
> _Why is the Woodbury Reformulation Effective?_ When considering the Woodbury reformulation in Eq (20), the correlation term of the re-derived conditional probability distribution is equivalent to $\mathbf{P}_1 \mathbf{S}_0^{-1} \mathbf{P}_1^\top + \mathbf{K}_1$. Since $\mathbf{S}_0^{-1}$ is positive definite and symmetric (thus allowing Cholesky decomposition), $\mathbf{P}_1 \mathbf{S}_0^{-1} \mathbf{P}_1^\top + \mathbf{K}_1$ can be re-derived into the following low-rank form:
> $\mathbf{P}_1 \mathbf{S}_0^{-1} \mathbf{P}_1^\top + \mathbf{K}_1 = \mathbf{P}_1 (\mathbf{LL}^\top) \mathbf{P}_1^{\top} + \mathbf{K}_1 = (\mathbf{P}_1\mathbf{L})(\mathbf{P}_1\mathbf{L})^\top + \mathbf{K}_1.$
>
> _The necessity of the Low-Rank Gaussian._ While the Woodbury reformulation ensures the fast inversion of $\mathbf{S}_{0}$, without the Low-Rank Gaussian implementation, the model would still be required to instantiate the dense $n \times n$ covariance matrix of the conditional distribution. This results in out-of-memory errors. Thus, treating the distribution as a Low-Rank Gaussian is imperative to fully realize the scalability advantages of the Woodbury reformulation.
>
> To summarize, the Woodbury reformulation improves efficiency by transforming the distribution used in direct inference into a low-rank multivariate Gaussian through appropriate mathematical derivation. We appreciate the reviewer’s insightful comment, as it has allowed us to enhance the completeness of our paper. We have incorporated these details into Appendix C.
>
> ---
>
> __(W2) The proposed model uses SNEA as their backbone and show the improvements in computational efficiency. However, from practical point, considering that the best prediction performances are achieved among TrustSGCN, SLGNN, further computational comparison in the Appendix can be added in Section 5. In some sense, the achievement (437 times faster convergence than baseline) can be viewed as an overstatement considering that the baseline is SNEA.__
>
> Following the reviewer’s suggestion, we relocated the full computational comparison to Tab. 3 in Section 5.1, and _removed_ our statement of 437 times faster convergence.

---

> ### Author Response · Authors · 2025-11-23
> **Official Comment for bz2b (2/2)**
>
> __(Q1) The notion of missing edges can be further clarified. In conventional GNN or in graph learning models, missing edges refer to unobserved relationships.__
>
> We thank the reviewer for clarifying this. The main reason for this mismatch is due to the difference in the target task. In the link sign prediction task we focus on, the edge's existence is already assumed, and the goal is to classify its sign (+/-). Therefore, our definition of an "unseen" or "missing" edge is closer to an “unlabeled” edge; that is, an "edge whose sign is unknown” under this specific setting. As the reviewer correctly noted, these unlabeled edges are not used during training. The model learns node embeddings and a projection layer exclusively using training edges with their known signs. At inference, an unseen test edge is presented, and the model constructs its embedding from the learned node embeddings first. Then, the trained projection layer estimates the parameters ($a$, $t$) of the relaxed Bernoulli distribution, which determines the sign of the test edge. We revised our manuscript to make this clearer. (Please refer to "Generalization to Unobserved Edges" paragraph in Section 3.2., line [250 - 256])
>
> ---
>
> __(Q2) I wonder how their ‘label softening’ is different from ‘label smoothing’ which is a well-established term.__
>
> We thank the reviewer for this clarification as well. We initially used the new term "label softening" to avoid potential ambiguity with the mathematical "smoothness" in our analysis, but we now agree that following the established one would be clearer and this will not confuse readers as much as we worried. We corrected this term and added an appropriate citation throughout the revised manuscript.
>
>
> ---
>
> __(Q3) Can you further elaborate why the correlation matrix should satisfy the condition in line 180-181? It is unclear why this condition should be added supposedly matrix R is the same from line 127-128 which is directly borrowed from CopulaGNN.__
>
> As shown in Lines [127-128], the Gaussian Copula Density equation includes the term $1 / \sqrt{\det {\mathbf{R}}}$, which mathematically requires the correlation matrix $\mathbf{R}$ to be positive-definite. This fundamental constraint is identical for both our model and CopulaGNN. The methodological difference lies in _how_ this positive definiteness is achieved. CopulaGNN adopts the method from [1], which derives a precision matrix using the Graph Laplacian and then inverts it. However, this approach is intractable for our edge-centric formulation because it necessitates constructing a line graph, which is not only computationally intensive but also does not inherently yield the low-rank structure required for our efficiency goals. This would preclude the use of the Low-Rank Gaussian implementation, leading to prohibitive computational costs. To address this while prioritizing scalability, we adopt an alternative strategy: ensuring the positive-definite condition by adding an epsilon identity term to the Gramian of the edges, which simplifies the problem and satisfies the constraints without compromising efficiency.
>
> ---
>
> [1] Jia and Benson. Residual correlation in graph neural network regression. (KDD 2020)

---

### Author Response · Authors · 2025-12-02
**Rebuttal Summary for Area Chair**

We are deeply grateful to the Area Chair for managing our submission during these exceptional circumstances. We have summarized the reviewers' questions and concerns into the following **five key points**.

---

**1. Prediction accuracy is not always superior than previous SOTA methods. (6GuR W1)**

We clarify that our primary goal is not on achieving a better prediction accuracy, but more on significantly superior computational efficiency and scalability. We argue that this is particularly important given that existing methods suffer from prohibitive memory usage and slow convergence, limiting their real-world applicability.

---

**2. It is unclear if the proposed method is robust to the choice of the backbone encoder, since it is experimented only with SNEA. (6GuR Q3, y8s8 Q1)**

We verified the robustness of our method by additional experiments with a 3-layer MLP and SLGNN (previous SOTA), as suggested by reviewers.

---

**3. Provide more intuitive and tangible evidence showing the effectiveness and stability of the proposed method. (bz2b W1, 6GuR W4, 87t8 Q1)**

In response to the reviewers’ suggestion, we visually analyze what our model has learned (Appendix F.2). We elaborate on the low-rank Gaussian to clarify scalability (Appendix C), and also provide condition numbers to verify numerical stability.

---

**4. The current manuscript describes theoretical analysis excessively; it would be nicer to have more experimental discussion in the main paper. (6GuR W5, y8s8 W4/W6, 87t8 W1)**

Reflecting this feedback by multiple reviewers, we rebalanced the contents by making theoretical analysis more concise (Sec. 4) and supplementing more experimental results (Sec. 5) from Appendix.

---

**5. Hyperparameter studies on η and ϵ are insufficient. (y8s8 W3, 87t8 Q2)**

Following the reviewer's suggestion, we significantly expand the hyperparameter search space and incorporate additional experiments focusing on both performance and convergence (Sec. 5.4).

---

### Meta-Review · Area_Chair_P6DG · 2026-01-14

**Summary:**

This paper provides a framework for link sign prediction in signed graphs. It computes correlation between two edges that are connected via one node. At a high level, they cast correlation matrix as the Grammian of edge embeddings, which ultimately reduces the number of parameters. All the reviewers are generally positive about the work. I think similar works were already explored in literature (https://jmlr.csail.mit.edu/papers/volume9/airoldi08a/airoldi08a.pdf). I think it would be good to compare their method with a probabilistic model where edge is modeled as joint distribution of latent functions of node features or even community membership. In this context, the authors can also explore stochastic block modeling (SBM). While such techniques are developed in pre-deep learning time, they are quite useful in practice.

However, since the reviewers already were moderately positive, I vote for an accept.

**Reviewer Concerns:**

The authors addressed the concerns of the reviewers.

**Reviewer Scores:**

Probably, they would have kept the same score or one of the them could increase. I found the reviewers to be somewhat non-committal in this case.

---

### Decision · Program_Chairs · 2026-01-26

Accept (Poster)